# Sun-Induced Fluorescence as a Proxy of Primary Productivity across Vegetation Types and Climates

Mark Pickering [1], Alessandro Cescatti [2], and Gregory Duveiller [2,3]

[1]JRC consultant, Ispra, Italy
[2]European Commission, Joint Research Centre, Ispra, Italy
[3]Max Planck Institute for Biogeochemistry, Jena, Germany

**Correspondence:** Mark.Pickering1@ext.ec.europa.eu, Alessandro.Cescatti@ec.europa.eu, GDuveiller@bgc-jena.mpg.de

**Abstract.** Sun-induced chlorophyll fluorescence (SIF) retrieved from satellites has shown potential as a remote sensing proxy for gross primary productivity (GPP). However, to fully exploit the potential of this signal, the robustness and stability of the SIF-GPP relationship across vegetation types and climates must be assessed. For this purpose, current studies have been limited by the availability of SIF datasets with sufficient spatial resolution to disentangle the signal between different vegetation cover types. To overcome this limitation the analysis uses GOME-2 (Global Ozone Monitoring Experiment 2) SIF retrievals, downscaled to a resolution of $0.05°$ ($\sim$ 5km) to explore the relationship between SIF and FLUXCOM GPP ($GPP_{FX}$), a data-driven dataset of primary productivity obtained by upscaling flux-tower measurements. The high resolution of the downscaled SIF ($SIF_{DS}$) dataset allows the relationships to be broken down by vegetation cover for separate climate zones, thus enabling a confrontation between GPP and SIF at fine granularity. This analysis first investigates the spatial and temporal relationships between FLUXCOM GPP and downscaled SIF at a global scale. A reasonably strong linear relationship is generally observed between $SIF_{DS}$ and $GPP_{FX}$ in all vegetation categories, and an analysis of covariance (ANCOVA) shows that the spatial response is similar between certain plant traits, with some distinction between herbaceous and woody vegetation, and notable exceptions, such as equatorial broadleaf forests. Geographical regions of non-linearity suggest where $SIF_{DS}$ could potentially provide information about ecosystem dynamics that are not represented in the FLUXCOM GPP dataset. With the demonstration of downscaled SIF as a proxy for GPP, the response of $SIF_{DS}$ to short-term fluctuations in several meteorological variables is analysed and the most significant short-term environmental driving and limiting meteorological variables determined. Vegetation groupings of similar SIF-meteo response reinforce the vegetation categorisations suggested by the ANCOVA analysis. This comparative exploration of two of the most recent products in carbon productivity estimation shows the value in downscaling SIF data, provides an independent probe of the FLUXCOM GPP model, enhances our understanding of the global SIF-GPP spatio-temporal relationship with a particular focus on the role of vegetation cover, and explores the similarity of the SIF and GPP responses to meteorological fluctuations. Additional analyses with alternative SIF and GPP datasets support these conclusions.

# 1 Introduction

Accurately quantifying the gross primary productivity (GPP) of vegetation systems across the globe is vital for modelling the future trajectories of atmospheric carbon fluxes and making projections regarding the Earth's climate. Indeed one of the largest sources of uncertainty in the carbon cycle is represented by the interaction between atmospheric carbon dioxide, climate and terrestrial ecosystem dynamics (Friedlingstein et al., 2019; Anav et al., 2015). Photosynthesis drives this interaction, with vegetation removing carbon from the atmosphere and investing it in growth, cell maintenance and respiration. In turn, photosynthesis is regulated by environmental conditions, and, as climates change, both the mean weather and its variability will change, impacting the productivity of vegetation systems (Seneviratne et al., 2012).

It is not possible to directly measure GPP at a global level, however many techniques have been developed to derive productivity at different scales using a range of data-driven or model-based approaches. Light use efficiency (LUE) models, for example, estimate GPP as a function of the absorbed photosynthetically active radiation (APAR), the efficiency of utilising light in photosynthesis $\epsilon_{LUE}$ and the effect of climatic constraints, such as temperature (T) and precipitation (P):

$$\text{GPP} = \epsilon_{LUE} \text{ x APAR x } f(\text{T}) \text{ x } f(\text{P}) \tag{1}$$

(Ryu et al., 2019; Running et al., 2004; Zhang et al., 2017; Lee et al., 2013; Pei et al., 2022).

A relevant assessment based on a process-oriented ensemble, known as TRENDY, provides a model-based estimation of global GPP ranging between $83 - 172$ PgC yr$^{-1}$, with the wide range of values highly dependent on the model assumptions. Eddy covariance sites, or flux towers, provide the most accurate ways of measuring carbon fluxes at ecosystem scale, through the systematic observation of the net ecosystem exchange of $CO_2$. These measurements have been standardised and made available thanks to the FLUXNET initiative that is linking different continental networks of eddy covariance towers (Baldocchi et al., 2001). The FLUXCOM project has upscaled FLUXNET data to a global estimate of GPP using machine learning methods to integrate site-level observations, satellite remote sensing information, and meteorological data (Tramontana et al., 2016). Whilst FLUXCOM is a large step forward in estimating GPP at a global level, it is not without its limitations and uncertainties. In fact, the various FLUXCOM GPP estimates use an ensemble of different machine learning methods and data inputs, which result in a broad spread of mean global GPP estimates among the ensemble members between $108 - 130$ PgC yr$^{-1}$. A comparative study between FLUXCOM and TRENDY finds that for $70\%$ of the globe at least the 9 out of 16 TRENDY models fall outside the FLUXCOM range (Jung et al., 2020).

In recent years, sun-induced chlorophyll-$a$ fluorescence (SIF), retrieved from space-based instruments, has grown in use as a remotely sensed proxy for GPP, in addition to more traditional remote proxies such as spectral vegetation indices (Frankenberg et al., 2011a; Joiner et al., 2011; Porcar-Castell et al., 2014). This fluorescent light - resulting from the re-emission by leaves of incident photons at lower energy - is considered to be the mechanism developed by plants to respond near-instantaneously to rapid perturbations in the environmental conditions of light and temperature, with the SIF yield also dependent on biophysical conditions such as the concentration of the $CO_2$-fixing enzyme Rubisco and drought stress (Frankenberg and Berry, 2017; Ryu et al., 2019). The SIF flux can similarly be expressed in terms of the absorbed incident radiation and the efficiency with which

this radiation is converted into fluorescent radiation, $\epsilon_F$:

$$SIF = \epsilon_F \text{ x } \epsilon_{esc} \text{ x APAR} \tag{2}$$

where the term, $\epsilon_{esc}$, accounts for the efficiency of photons to escape re-absorption and scattering by other leaves in the canopy (Lee et al., 2013). Rearranging the equations for instantaneous SIF and GPP fluxes:

$$GPP = \frac{\epsilon_{LUE}}{\epsilon_F \text{ x } \epsilon_{esc}} \text{ x SIF} \tag{3}$$


we see that under conditions in which the various conversion efficiencies remain constant, there is a linear relationship between SIF and GPP. Whilst at small spatio-temporal timescales, where leaf chemistry is particularly sensitive to changes in absorbed photosynthetically active radiation and the fraction of fluoresced photons escaping from the canopy, there is evidence for the divergence of SIF and GPP from linearity, it appears that the broader canopy-scale relationship smooths over these non-

linearities (Magney et al., 2020). Indeed, there is a substantial body of evidence that shows that SIF, measured from space-based instruments, is positively correlated with leaf photochemistry, often exhibiting a generally linear relationship in both space and time, and across spatio-temporal scales (Zhang et al., 2016; Sun et al., 2018; Magney et al., 2020). However, this SIF-GPP relationship may exhibit some dependency on the vegetation type, for example through the canopy structure that is affecting $\epsilon_{esc}$, as well as the leaf photochemical properties and external conditions, for example climate drivers. Due to the relatively fast

response of SIF and close link to leaf photochemistry, compared to other remote indicators of greenness, such as NDVI, SIF has the potential to be an indicator of environmental stress for the plant photosystem (Walther et al., 2019; Jiao et al., 2019).

There is currently no orbiting satellite designed explicitly to directly measure SIF from space. The first that will do so is the exploratory mission FLEX, scheduled for launch in the coming years (Coppo et al., 2017). In the meanwhile, SIF has been retrieved from other instruments designed for measuring the atmosphere greenhouse gas concentration, namely GOSAT,

SCIAMACHY, the Global Ozone Monitoring Experiment-2 (GOME-2), the Orbiting Carbon Observatory 2 (OCO-2) and the TROPO-spheric Monitoring Instrument (TROPOMI) (Guanter et al., 2012; Joiner et al., 2012, 2013; Sun et al., 2018; Köhler et al., 2018b; Guanter et al., 2021; Doughty et al., 2019). However, several issues hamper the use of these data for the quantification of terrestrial GPP. First, some instruments (GOSAT, OCO-2) are sampling the surface, leaving wide gaps between different satellite overpasses. Second, the time series of observations is shorter than desired for carbon science, especially for

the more recent instruments (e.g. OCO-2 and TROPOMI). Third, most have a spatial resolution that is too coarse to isolate homogeneous vegetation patches of distinct land cover types.

Efforts have been made to improve the resolution and coverage of SIF datasets by combining SIF data with other high resolution remote sensing data (Gentine and Alemohammad, 2018; Li and Xiao, 2019; Zhang et al., 2018a; Yu et al., 2018; Gensheimer et al., 2022). These approaches generally rely on statistical inference, through machine learning methods. A down-

scaling methodology, based on a light use efficiency model, combines the GOME-2 data with several explanatory biophysical variables in a process oriented scheme. The resulting dataset has a spatial resolution of $0.05°$ (5km) and is therefore at a scale relevant to studies of land cover at global scale (Duveiller et al., 2020; Duveiller and Cescatti, 2016). This model ensures that the downscaling method is grounded in theory whilst also preserving the GOME-2 signal. Downscaling the SIF in this way

results in a high resolution dataset with a reasonably long archive, improving accuracy in the exploration of the SIF relationship with vegetation cover.

If downscaled sun-induced fluorescence is to be used as a proxy for ecosystem productivity it is important to understand the spatial and temporal relationships between SIF and the current state-of-the-art GPP datasets at a global scale, and in particular understand how they deviate for differing vegetation covers and climate zones. To this end, this paper serves several purposes. Firstly, the analysis provides a thorough test of the utility of the downscaling method to reproduce known SIF-GPP patterns, in particular through the spatio-temporal correlation between downscaled SIF and FLUXCOM GPP. Exploring variations in the FLUXCOM GPP with an independent SIF dataset, often likewise regarded as a proxy to GPP, helps to probe its strengths and limitations through areas of coherence and divergence. Similarly, comparisons with alternative SIF and GPP products such as TROPOMI SIF (Guanter et al., 2021) and FluxSat GPP (Joiner and Yoshida, 2021) are provided in an appendix, in order to ensure the consistency and robustness of the conclusions. Second, as a global, high-resolution investigation into the SIF-GPP relationship, the analysis allows us to learn more about the differing spatial linear relationship between SIF and GPP and their variation in nature with a particular focus on similarities and differences between vegetation covers. This allows the determination of which vegetation covers have a similar SIF-GPP response, and for which vegetation covers care should be taken in the use of SIF as a proxy for GPP. Finally, having established the spatio-temporal relationship between the downscaled SIF and the FLUXCOM GPP, the paper investigates the response of downscaled SIF to fluctuations in several meteorological factors, in the process determining the most significant driving and limiting meteorological factors in monthly SIF fluctuations. By utilising the high resolution of the downscaled SIF, it is possible to understand with improved confidence the extent to which vegetation cover plays a role in these relationships using dedicated techniques (e.g. Álvaro Moreno-Martínez et al., 2018).

## 2  Data

### 2.1  Vegetation cover data

The data relating to the vegetation cover of each pixel is derived from the Copernicus Climate Change Service (C3S) via the climate data store platform, with the data created by the ESA CCI program (CCI, 2017; Defourny, 2019). The land cover classes are converted to vegetation covers, as used by dynamic global vegetation models, whilst aggregating the data to a spatial resolution of $0.05°$. The following vegetation covers are considered: grassland = 'GRA', crops = 'CRO', evergreen broad-leaf forest = 'EBF', deciduous broad-leaf forest = 'DBF', evergreen needle-leaf forest = 'ENF', and deciduous needle-leaf forest = 'DNF'. To ensure a high homogeneity in the selected data, the dominant vegetation type must cover at least $75\%$ of a pixel and with no change in the majority land cover classification over the considered years, 2007-2014.

### 2.2  Climate classification

The climate zone classification used in the analysis follows the Köppen-Geiger climate classification scheme (Kottek et al., 2006; Rubel and Kottek, 2010; Rubel et al., 2017). The classification maps are representative of the period 1986-2010 and are

available at a spatial resolution of $0.0833°$ which are extrapolated via binomial interpolation to grid cells (referred to hereon as pixels) of $0.05°$.

Four broad categories are considered from this scheme: equatorial, arid, temperate and continental. Equatorial contains 'Group A' climate regions: areas where each month is above $18°C$ and with high precipitation. Arid regions are 'Group B' climates: areas defined by low precipitation. Temperate regions are 'Group C' climates: with the coldest month averaging

$0 - 18°C$ and at least one month averaging more than $10°C$. Finally, continental regions are 'Group D' climates: at least one month must average below $0°C$ and at least one month above $10°C$. Figure 1 shows the spatial distribution of the global climate groupings and the dominant vegetation cover of the pixels considered in the analysis.

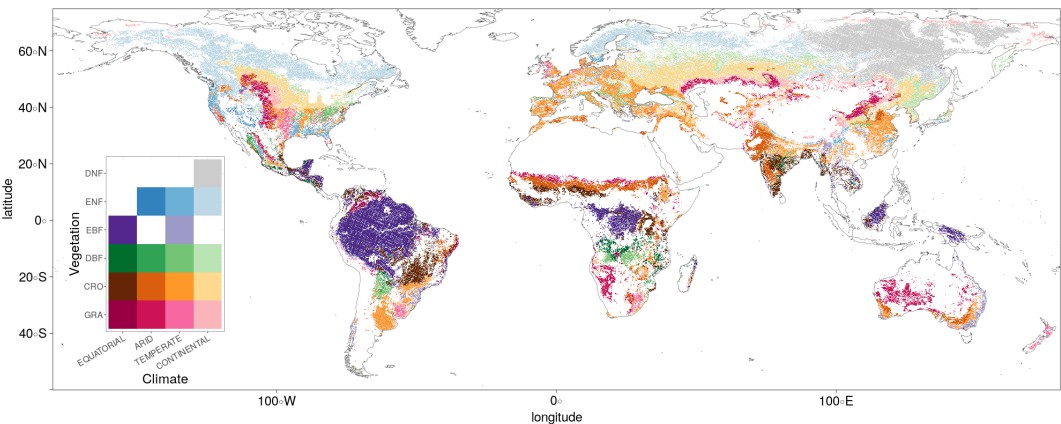

**Figure 1.** The dominant Köppen-Geiger climate zone and vegetation cover corresponding to each of the pixels passing the full set of selection requirements.

## 2.3  Growing season data

The Vegetation Index and Phenology (VIP) global dataset from NASA's Making Earth System Data Records for Use in Re-

search Environments (MEaSUREs) program is used to define the growing seasons at each grid cell for each year (Didan, 2016). The datasets are created using surface reflectance data from the MODIS instrument. This data provides a consistent NDVI and EVI measurement from which to characterise the vegetation phenology. The Vegetation Index and Phenology (VIP) Phenology NDVI (VIPPHEN) v004 dataset has a global spatial resolution of $0.05°$ and provides annual metrics on the start and length of the growing season for each pixel for the years 2000-2014.

Whilst correlation between SIF and GPP has been observed across all seasons, only the relationship between downscaled SIF and FLUXCOM GPP during the growing season of each pixel is considered in the present study (Magney et al., 2019; Bowling et al., 2018). This removes the effect of winter periods, when there is little primary productivity and when the retrieval of SIF can be problematic at northern latitudes. Off-season, the relatively weak SIF signal and the quality requirements in the downscaling process result in a dataset with gaps. Including this data in the analysis would likely result in distorted conclusions

regarding average downscaled SIF signals over the time period. Additionally, only the first growing season of each year is considered in regions with multiple growing seasons.

## 2.4 SIF data

Two SIF datasets are considered in this analysis, produced via the downscaling method detailed in references Duveiller and Cescatti (2016) and Duveiller et al. (2020). The two retrievals have a spectral wavelength around 740nm, and differ in the retrieval method for obtaining the input data from the GOME-2 satellite, the first product developed by Joiner et al. (2013), is referred to as SIFJJ in this document, whilst the second, developed by Köhler et al. (2015), is referred to as SIFPK. A correction factor to convert the instantaneous SIF to the daily average is applied to both datasets to ensure comparability with estimates at different acquisition times (Frankenberg et al., 2011b; Köhler et al., 2018a). The downscaling method calibrates these input retrievals via a light use efficiency model using high resolution biophysical variables from the MODIS (MOderate Resolution Imaging Spectroradiometer) instrument of the Terra and Aqua Satellites. The optimal combination of variables is identified in combination with OCO-2 data, and the downscaled dataset is found to have a high level high spatio-temporal agreement with observations from the TROPOMI mission.

The resulting downscaled SIFPK and SIFJJ products have a spatial resolution of $0.05°$ and a temporal separation of 8 days (with measurements averaged over a sliding window of 16 days). The datasets currently cover the timespan 2007-2017, with 46 measurements each year (with the exception of the 2007 SIF dataset, containing 44). Duveiller et al. (2020) shows that the downscaled SIFJJ dataset is found to have a slightly higher level of agreement with the OCO-2 validation data than the downscaled SIFPK dataset and so is primarily used in this paper, and is henceforth referred to as 'downscaled SIF' (or $SIF_{DS}$). The higher agreement likely results from the spatial smoothing step of the downscaling process that benefited the noisier SIFJJ more than the SIFPK.

To ensure high quality in the data, and compatibility with the other datasets, several requirements are placed on each pixel in each year, further to the requirements detailed in Duveiller et al. (2020), Köhler et al. (2015), and Joiner et al. (2013). There must be at least 10 instances of valid $SIF_{DS}$ observations of the pixel within the growing season with fewer than $40\%$ of the expected number of $SIF_{DS}$ missing or invalid. There must also be least six years of valid measurements satisfying the requirements between 2007-2014. The selections ensure that the SIF signal, which is relatively weak compared to background noise, and affected by cloud coverage, is representative of the growing season as a whole as well as excluding regions with short growing seasons that may be more susceptible to fluctuations from unusual weather conditions. Requiring multiple years of data passing the quality requirements enables the investigation of temporal trends, whilst also ensuring that the measurements are representative of each pixel.

In order to reduce spatial auto-correlation and the double-counting of interpolated pixels in other datasets, pixels considered in the analysis must be separated by a two-pixel gap in all directions (Ploton et al., 2020). Each pixel is matched with the dominant vegetation cover and climate classification, as well as FLUXCOM GPP and meteorological data, passing the respective requirements. Figure 2 shows the mean downscaled SIF for the growing season of each pixel passing the analysis selection requirements, averaged over the period 2007-2014.

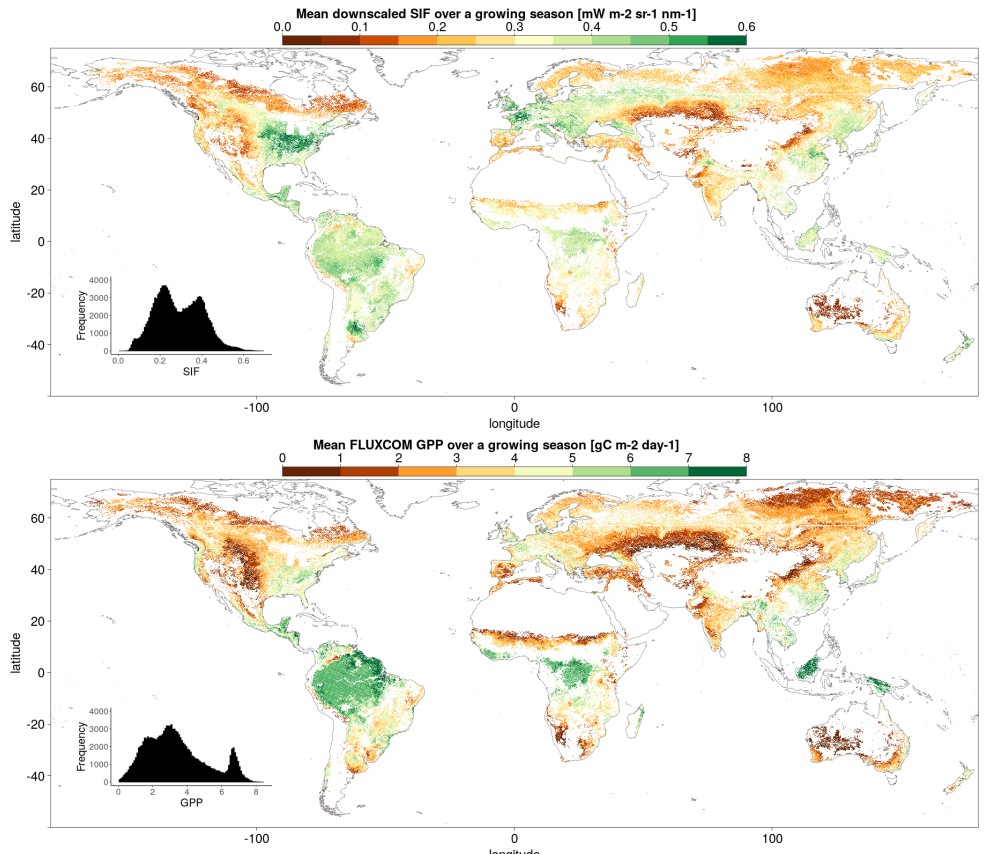

**Figure 2.** Mean downscaled SIF (above) and FLUXCOM GPP (below) over the growing season, corresponding to each of the pixels passing the full set of selection requirements. The SIF$_{DS}$ fluxes and GPP$_{FX}$ estimations for each pixel are averaged over multiple years between 2007 and 2014.

### 2.5 GPP data

The gross primary productivity (GPP) dataset is provided by the FLUXCOM project, measured as a daily carbon uptake [gC m$^{-2}$ day$^{-1}$] (Jung and FLUXCOM Team, 2016; Tramontana et al., 2016; Jung et al., 2020). In the 'RS only' setup used in this analysis and described in Tramontana et al. (2016) and Jung et al. (2019), an ensemble of nine machine learning methods merge carbon flux estimations from FLUXNET eddy covariance towers with remote sensing data taken or derived from the MODIS sensor to estimate gross primary productivity across the terrestrial surface. The remotely sensed data includes land

surface temperature, fraction of absorbed photosynthetic active radiation, normalized difference vegetation index, normalized difference water index and land surface water index. The resulting dataset, hereon referred to as 'FLUXCOM GPP' (or GPP$_{FX}$), consists of a global estimate of GPP at a spatial resolution of $0.0833°$. These estimates occur in timesteps of 8 days (46 over the course of a year) and cover the downscaled SIF data collection period up until the year 2016.

The GPP pixels are extrapolated via binomial interpolation to $0.05°$ pixels in order to focus on the comparison with the $\text{SIF}_{DS}$ pixels. Figure 2 shows the FLUXCOM GPP for the growing season of each pixel passing the analysis selection requirements, averaged over the period 2007-2014.

## 2.6 Meteorological data

ERA5 is the fifth generation ECMWF reanalysis global climate and weather dataset, and the ERA5-Land dataset replays the land component of the reanalysis to provide land variables at an enhanced resolution at $0.1°$. The dataset is extrapolated via binomial interpolation to $0.05°$ pixels in order to focus on the comparison with the $\text{SIF}_{DS}$ pixels, with only non-consecutive months considered, in order to reduce temporal autocorrelation.

Meteorological variables are obtained from the ERA5-Land monthly reanalysis dataset (Muñoz Sabater, 2019b; Muñoz Sabater et al., 2021). These include: air temperature (t2m [°C]: temperature of air at 2m), surface net solar radiation (ssr [J $m^{-2}$]: amount of solar radiation reaching the surface of the Earth minus the amount reflected by the Earth's surface) and soil moisture (swvl1 [$m^3$ $m^{-3}$]: volume of water in soil layer 1, 0-7 cm, of the ECMWF Integrated Forecasting System). A variable that is not available is the mean monthly vapour pressure deficit (VPD [kPa]), the difference between the saturated vapour pressure and the actual vapour pressure (Grossiord et al., 2020). It is important in regulating the stomatal conductance of plants, and thus useful to relate to both SIF and GPP. Due to non-linearity in the vapour pressure-temperature response, the average saturated vapour pressure of each month is calculated from the average of the saturation vapour pressure at the mean daily maximum and mean daily minimum air temperatures over the course of the month, using the following formula (Allan and Pereira, 1998):

$$e_s = [e°(\text{T}_{max}) + e°(\text{T}_{min})]/2 \text{ where: } e°(\text{T}) = 0.061 \text{ x exp }^{17.27\text{T}/(\text{T}+237.3)} \tag{4}$$

The latter formula is also used in the calculation of the actual vapour pressure from the dewpoint temperature. The minimum and maximum air temperatures and the dewpoint temperature are taken from the ERA5-Land hourly reanalysis dataset (Muñoz Sabater, 2019a).

## 3 Methodology

The $\text{SIF}_{DS}$-$\text{GPP}_{FX}$ spatio-temporal relationship at global scale is analysed via several diagnostics. Linear models and analysis of covariance (ANCOVA) are performed to determine the similarities and dissimilarities in the response across different vegetation covers. Finally, the response of the $\text{SIF}_{DS}$ to fluctuations in meteorological conditions is investigated to assess the potential of this metric to diagnose the impact of environmental drivers. For the analysis, each $0.05°$ vegetated pixel is described by a time series of downscaled SIF, FLUXCOM GPP, and meteorological values, taken over the first growing season of each year between 2007 and 2014. This same set of 135,000 global pixels is used in each analysis of the current paper, with consideration given to the vegetation cover and climate zone of the pixels analysed.

Several sections of the analysis of the SIF-GPP spatio-temporal relationship are repeated with the alternative FluxSat GPP dataset (in place of the FLUXCOM GPP) and the TROPOMI SIF dataset (in place of the downscaled SIF) in order to ensure the robustness and consistency of the analysis. These can be found in appendix A3 and appendix A4 respectively.

## 3.1  The spatio-temporal relationship of SIF$_{DS}$ and GPP$_{FX}$

Since the processes and drivers of variability in SIF and GPP may differ in time and space, we designed an analytical framework to isolate the temporal components of the SIF$_{DS}$-GPP$_{FX}$ relationship at different temporal resolutions (intra-and inter-annual) from the spatial variations. The spatial component of the SIF$_{DS}$-GPP$_{FX}$ correlation is isolated by determining the multi-year mean SIF$_{DS}$ and mean GPP$_{FX}$ for each pixel. Here 'mean' refers to the mean daily value of the downscaled SIF or FLUXCOM GPP over the first growing season. These values are converted to a multi-year means by averaging over the period 2007-2014. The Pearson's spatial correlation coefficient, $r$, and a least-squares linear model are calculated at both a global scale, as well as over a local moving window of $2.5°$ for each climate-vegetation category, with the latter requiring at least 10 pixels within the moving window to be assessed and reported. The temporal component of the SIF$_{DS}$-GPP$_{FX}$ correlation and linear model is assessed at both the inter- and intra-annual scales. The inter-annual correlation, refers to the temporal relationship between the mean growing season SIF$_{DS}$ and GPP$_{FX}$ values between consecutive years at the same location. It should be noted that a temporal degradation in the GOME-2 instruments has been observed, potentially affecting the long-term analysis of SIF trends and therefore the SIF-GPP relationship, particularly from 2015 onwards (Zhang et al., 2018b). Whilst this may have a slight impact on the analysis presented here - which uses data collected up until 2014 - we nevertheless consider the inter-annual comparison of SIF$_{DS}$ and GPP$_{FX}$ worthwhile. Meanwhile, the intra-annual correlation refers to the relationship between individual SIF$_{DS}$ and GPP$_{FX}$ values made at 8-day timesteps within a growing season, in order to determine the internal growing season statistics. A minimum of 10 observations within a growing season over at least 6 years is required. The correlation and slope parameter of the least-squares linear relationship at each pixel is calculated for each year considered and averaged over the multi-year time period.

## 3.2  The spatial linear relationship between SIF$_{DS}$ and GPP$_{FX}$

The same process and data used to isolate and determine the spatial component of the correlation is used to determine the global spatial linear relationships between SIF$_{DS}$ and GPP$_{FX}$. For this purpose, an area-weighted least squares linear model fits the global SIF$_{DS}$-GPP$_{FX}$ distribution of pixels for each climate and vegetation cover. Whilst theoretically the leaf photosynthesis may be zero when the quantity of emitted SIF radiation is zero, this does not necessarily imply that the canopy level SIF-GPP relationship extends linearly to zero, as the canopy level SIF-GPP relationship smooths over known non-linearities at finer scales and lower SIF yields (Magney et al., 2020). Additionally, forcing the linear regression through the origin based on a prior expectation (in this case that SIF and GPP are simultaneously zero) that lies outside the bounds of the considered data will introduce a bias into the regression parameters. Therefore the intercept of the SIF$_{DS}$-GPP$_{FX}$ relationship is not forced through zero to account for this variation, as well as potential deviations from linearity in the sampled pixels.

### 3.3 Spatial analysis of covariance between SIF$_{DS}$ and GPP$_{FX}$

In order to assess and test similarities in the global SIF$_{DS}$-GPP$_{FX}$ response between vegetation covers, an ANCOVA (analysis of covariance) is performed. ANCOVA compares linear regressions between two or more groups whilst controlling for a covariate to test about the stastistical significance of the effects. In this specific case, the downscaled SIF covariate is controlled for in a spatial linear regression with the FLUXCOM GPP that differs between vegetation and climate groups. A comparison of the regression slope and intercept between pairs of vegetation cover groupings is conducted in terms of the significance (through the p-value) and the size of the effect (through $\eta^2$). The p-value for the slope parameter is the probability of obtaining an equal or more extreme difference in the regression slopes of two vegetation groups under the null hypothesis that the vegetation cover has no effect. The p-value for the intercept additionally assumes the null hypothesis for the regression slope. The size of the effect is measured through $\eta^2$ ($0 \leq \eta^2 \leq 1$), the proportion of the sum of squares from the nominal grouping of vegetation cover, $SS_{veg}$, to the overall sum of squares for the linear relationship, $SS_{lm}$:

$$\eta^2 = SS_{veg}/SS_{lm} \tag{5}$$

Therefore, $\eta^2$ gives the proportion of the variance attributable to the vegetation cover grouping and is conceptually similar to the significance of the coefficient of determination, $r^2$, in linear relationships. The p-value provides evidence for whether the difference in SIF$_{DS}$-GPP$_{FX}$ response is significant between vegetation covers, and $\eta^2$ can be thought of as the magnitude of that difference. For each climate grouping, pairwise ANCOVA comparisons are made between vegetation covers for a sample of 400-1000 pixels.

### 3.4 Estimating global GPP with downscaled SIF

The derived global spatial linear SIF$_{DS}$-GPP$_{FX}$ relationships are used to project the downscaled GOME-2 SIF into an estimate of gross primary productivity, GPP$_{Est}$. This is also interpreted in terms of absolute and percentage differences to the FLUXCOM GPP, with the percentage difference calculated as:

$$GPP_{diff} = 100 \text{ x } (GPP_{Est} - GPP_{FX})/GPP_{FX} \tag{6}$$

Mapping the differences between FLUXCOM GPP and GPP$_{Est}$, estimated using the downscaled SIF and SIF$_{DS}$-GPP$_{FX}$ relationships, enables the display of areas where the global, category-dependent, linear relationships succeed or fail in replicating the GPP$_{FX}$ from the local SIF$_{DS}$ observations. There are four different groupings of global linear relationships used in the breakdown. Firstly the GPP estimate depends only on separate SIF$_{DS}$-GPP$_{FX}$ relationships for each Köppen-Geiger climate zone; secondly, the GPP estimate depends is carried out separately for each vegetation cover, with no consideration for the climate zone; third, both the climate zone and vegetation cover are taken into account; and finally the groupings used are suggested from the analysis of covariance. The latter is also used to display an estimate of global GPP based on the downscaled SIF, scaled by the FLUXCOM GPP relationships.

### 3.5 The SIF$_{DS}$ response to meteorological fluctuations

The response of length-of-day corrected downscaled SIF to anomalies in a number of meteorological variables is analysed in order to determine similarities in response between different vegetation covers and to understand the driving meteorological factors for SIF fluctuations in different climate zones. A focus is given to meteorological extremes, investigated through the z-score from the long-term monthly mean. The study uses the same initial data as the investigation into SIF$_{DS}$-GPP$_{FX}$ response, however monthly averages of the SIF$_{DS}$ are taken in order to compare with the month-averaged meteorological variables. The meteorological factors considered are air temperature, solar radiation, soil moisture and vapour pressure deficit. Additionally, only non-consecutive months within a growing season are included, in order to reduce temporal autocorrelation.

For every pixel, the mean and standard deviation of the SIF$_{DS}$ and meteorological variables is calculated for each month over the period 2007-2014. These individual monthly values are re-expressed as a z-score for each pixel - i.e. the difference to the 2007-2014 monthly mean, standardised by the standard deviation. The FLUXCOM GPP is also included in the analysis, though, as noted, the FLUXCOM GPP product takes several remotely-sensed climatic variables as input and so is not independent of the meteorological drivers. The inclusion of the GPP product enables a comparison with the SIF$_{DS}$, giving insight into whether the SIF behaves as may be expected of an independent proxy for GPP.

## 4 Results

### 4.1 The spatio-temporal correlation of SIF$_{DS}$ and GPP$_{FX}$

The Pearson's correlation coefficient, $r$, between the downscaled SIF and FLUXCOM GPP is projected into figure 3, to display areas of high and low temporal and local spatial correlation, along with the slope parameter of a least-squares linear model between the two. Meanwhile, figure 4 displays the global spatial and average global temporal correlations for each vegetation cover and climate zone for comparative purposes.

The first thing to note is that at a global scale, prior to the breakdown into separate climate-vegetation cover categories, there is a reasonably strong correlation between the downscaled SIF and the FLUXCOM GPP. Between different climate-vegetation groupings, however, there is variety in the strength of the correlation. Whilst the breakdown of the relationship by either climate zone or vegetation cover separately provides extra information in comparison to no breakdown, greater variability is shown from a breakdown by both categories simultaneously, highlighting the value of the downscaled SIF dataset in assessing the relationship with GPP across vegetation categories in different climates. The slight variation in correlation across different vegetation covers suggests that, although there are more similarities than differences, there is value in breaking down the relationship by vegetation cover.

The spatial and temporal analyses show that downscaled SIF functions as a reasonable spatial and temporal proxy for GPP, across multiple timescales and vegetation covers. The figures show that regions and vegetation-climate categories with high correlation in one spatio-temporal analysis generally show high correlation in another analysis, suggesting that spatial and temporal correlation in the SIF$_{DS}$-GPP$_{FX}$ datasets are actually interlinked. The highest correlations are almost exclusively

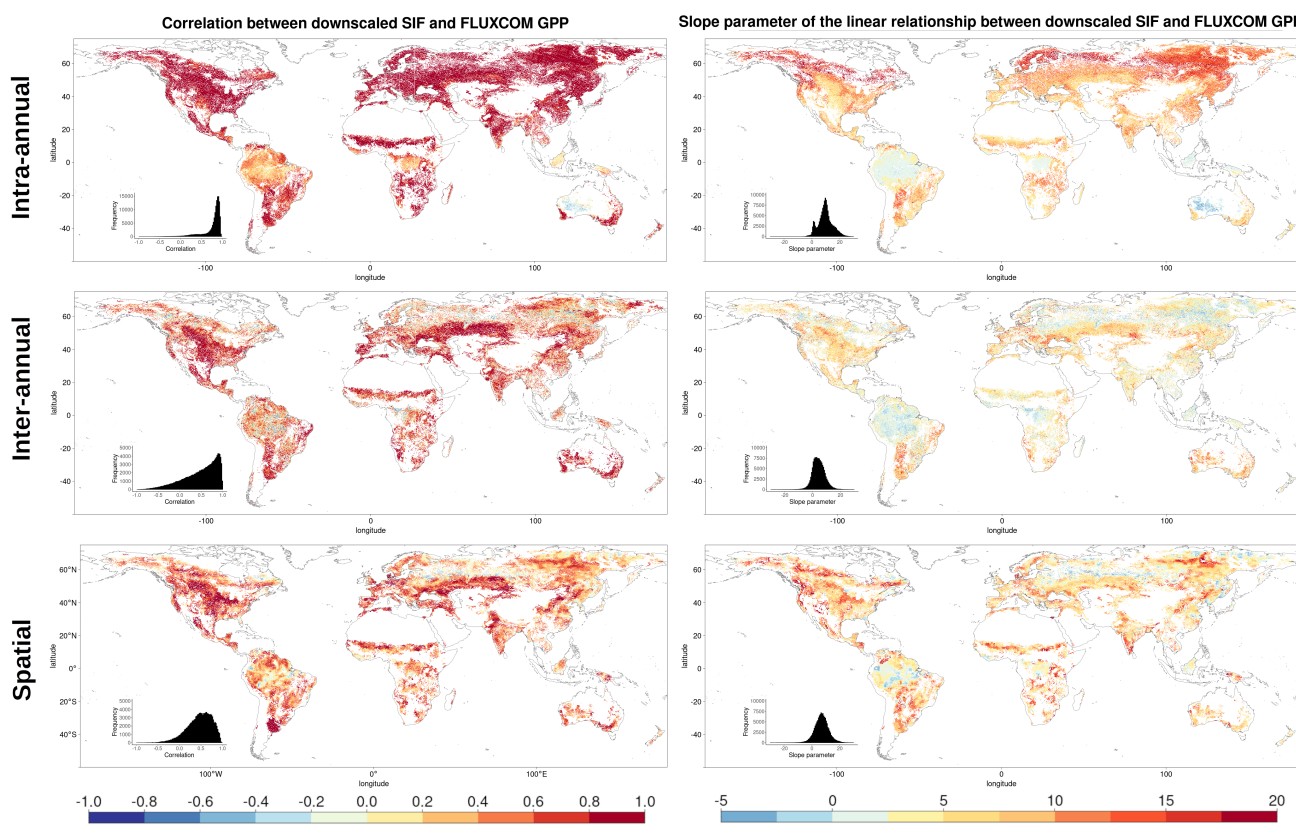

**Figure 3.** The Pearson's correlation coefficient, $r$, (left) and the linear model slope parameter (right) for: Above: the intra-annual temporal relationship between the downscaled SIF and FLUXCOM GPP over 8-day timesteps within a growing season. Middle: the inter-annual temporal relationship between the mean annual downscaled SIF and mean annual FLUXCOM GPP. Below: the spatial relationship between the mean annual downscaled SIF and mean annual FLUXCOM GPP, with the correlation determined for a given dominant vegetation cover and climate zone over a $2.5°$ moving window.

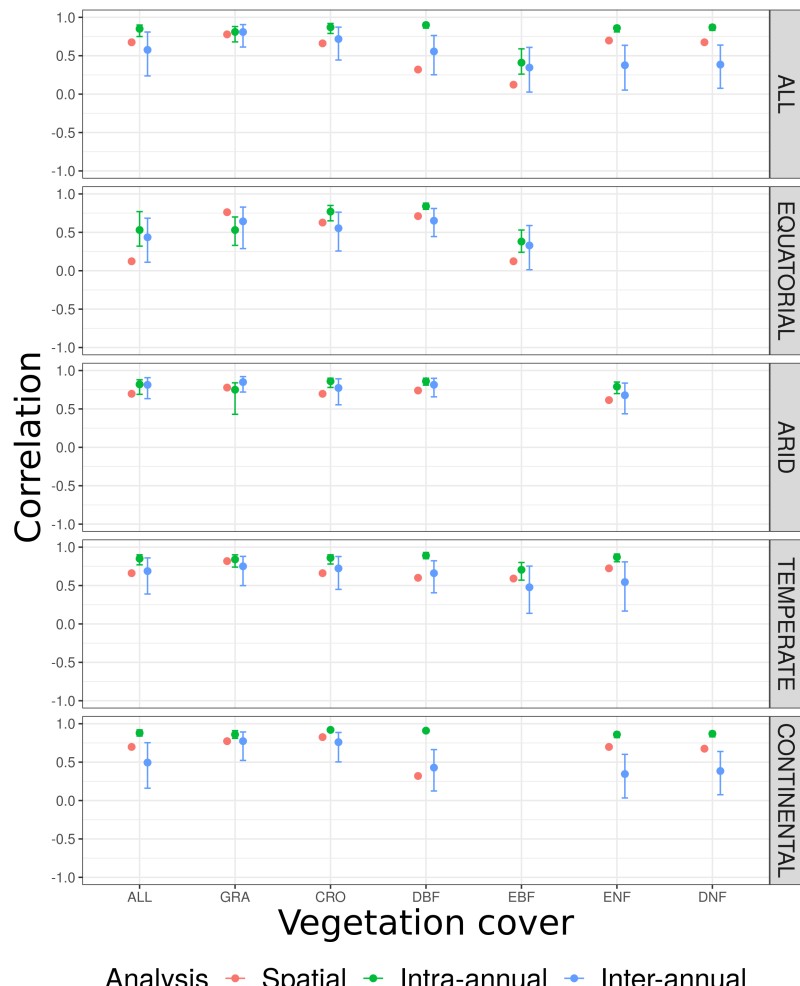

**Figure 4.** The Pearson's correlation coefficient, $r$, between downscaled SIF and FLUXCOM GPP for the spatial relationship, intra-annual temporal relationship within a growing season, and the inter-annual temporal trend across years. For the spatial analysis, a single global spatial correlation is calculated for each vegetation cover and climate zone, whilst the temporal relationships display the median global correlation and 25% upper and lower quantiles for each pixel, broken down by vegetation cover and climate zone.

found between between $SIF_{DS}$ and $GPP_{FX}$ within the same growing season as a result of the strong effect of seasonality in the key environmental drivers of primary productivity, such as radiation, temperature and water availability. Indeed, all vegetation-climate categories except for equatorial broadleaf forests exhibit $r > 0.5$, with all regions outside the tropics and the arid grasslands of central Australia showing high correlation. Larger intra-annual slope parameters between $SIF_{DS}$ and $GPP_{FX}$ are similarly found in the high latitude regions which experience the largest seasonality.

The spatial correlation and the temporal trend between years show similar features, though are generally weaker than the intra-annual correlation, with some regions of tropical rainforest and continental forest in Russia displaying anti-correlation.

There is also a wider distribution in the strength of the $SIF_{DS}$-$GPP_{FX}$ correlation. This is despite the fact that the temporal analyses have a more granular level of spatial detail, with each pixel more susceptible to fluctuations. This is particularly true of the inter-annual comparison, which uses fewer data points in the regression.

## 4.2 The spatial linear relationship between $SIF_{DS}$ and $GPP_{FX}$

Figure 5 shows the relative distribution and spatial linear relationship between the mean growing season FLUXCOM GPP as
a function of the respective mean values of the downscaled SIF during the growing season. The data are broken down into separate categories depending on the Köppen-Geiger climate grouping and dominant vegetation cover of the pixel.

The significant substructure in the $SIF_{DS}$-$GPP_{FX}$ distribution and greater deviation from the linearity in the 'ALL' categories, suggest that the $SIF_{DS}$-$GPP_{FX}$ spatial relationship response is dependent on both the climate and vegetation covers. There is also some evidence that there is a slight trend towards a reduction in the slope in cooler climates, though this may result from
325 factors other than the climate itself, for example, differences in the spatial distribution of vegetation between evergreen and deciduous types or between C3 and C4 crops and grasses.

In all categories except EBF, the spatial correlations are comparable to the relationship observed between FLUXCOM GPP and SIF measurements from the OCO-2 instrument, as seen in Sun et al. (2018) confirming the overall value of the downscaled SIF product for this specific exercise. In the Sun et al. (2018) study, the following correlation coefficients are exhibited (broken
down by biome): $r_{GRA} = 0.74$; $r_{CRO} = 0.88$; $r_{EBF} = 0.74$; $r_{DBF} = 0.8$; $r_{NF} = 0.84$ (needleleaf). The differences to this study may result from the selection criteria of the biomes, the singular grouping of vegetation covers across different climate zones, and the forcing of the linear relationship intercept through zero. In particular, the latter assumption of the SIF-GPP relationship leads to higher correlation coefficients compared to allowing the intercept to float. The observed linear relationship is found to be stronger with the FluxSat GPP dataset, as displayed in appendix A3
We acknowledge that, in some categories, a linear model may be too simplistic to represent the relationship between $SIF_{DS}$ and $GPP_{FX}$. This is more true for the woody plants which display some complexity in the $SIF_{DS}$-$GPP_{FX}$ relationship, in contrast to herbeaous vegetation, which remains highly linear, despite exhibiting a greater range in values. The clearest deviation from linearity is found in highly productive equatorial evergreen forests, where a wide range of spatio-temporal variation in $SIF_{DS}$ is observed, while a considerably smaller variability is reproduced in the modelled $GPP_{FX}$. This non-linearity is explored in
more depth in the discussion.

Whilst at first glance the heatmap of temperate deciduous broadleaf forests similarly hints at a plateau effect, the figure can in fact be divided into two areas of high $SIF_{DS}$ and low $SIF_{DS}$ data points corresponding to separate spatial locations. The lower $SIF_{DS}$ values correspond to deciduous forests in Southern Africa and South America, whilst the higher $SIF_{DS}$ values occur in North America and Europe, suggesting that there may not be global universality in the $SIF_{DS}$-$GPP_{FX}$ relationship, or that
different types of deciduous broadleaf forests found in distinct regions could respond differently, possibly based on differences in species composition. It should be noted that this distinction is not observed in the TROPOMI SIF dataset for the year 2020 (appendix A4).

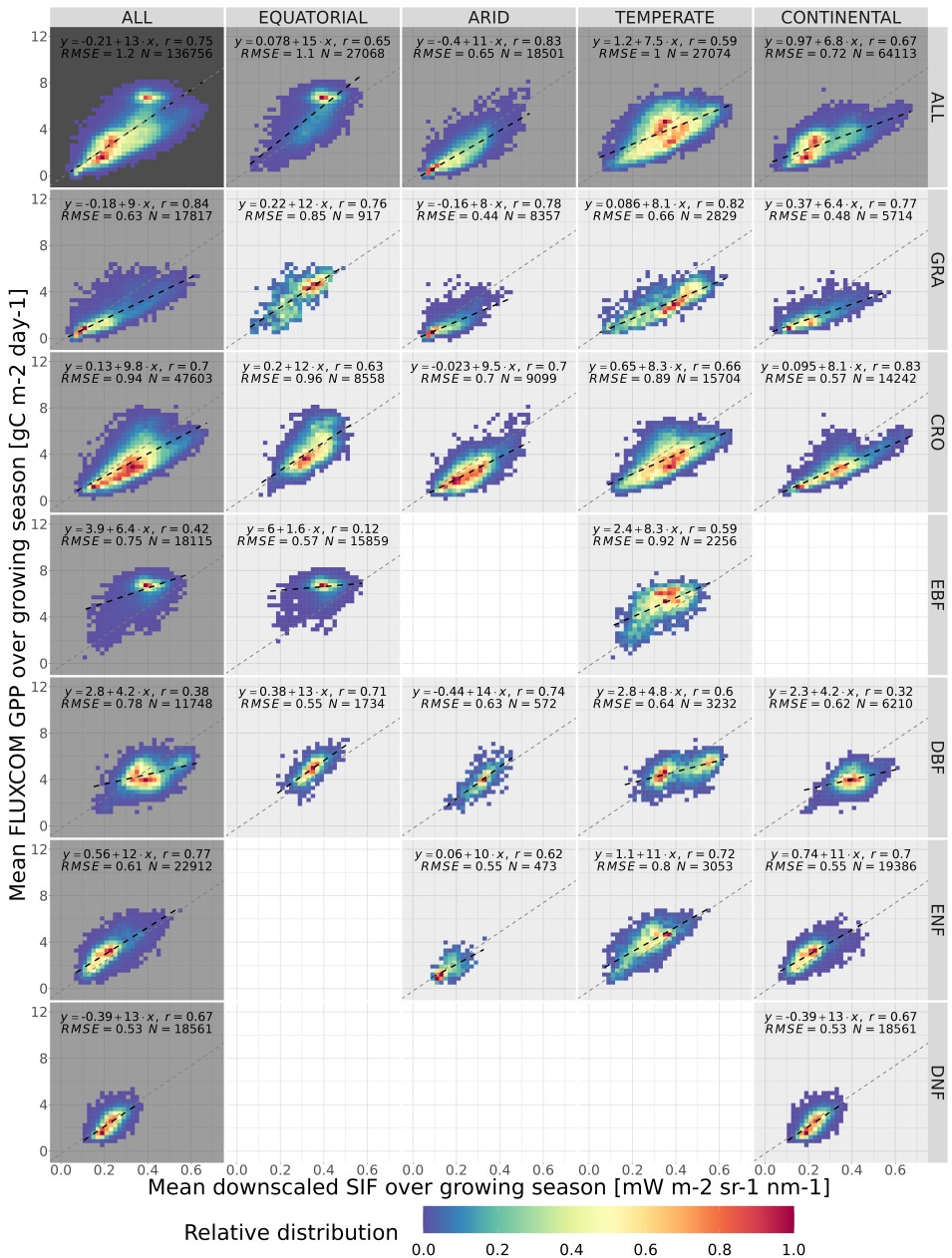

**Figure 5.** The spatial relationship between the mean growing season downscaled SIF and FLUXCOM GPP, broken down into separate Köppen-Geiger climate zones and vegetation cover categories. The plot shows the frequency distribution of pixels into $SIF_{DS}$-$GPP_{FX}$ bins, relative to the highest frequency bin in that category. A black dashed line representing a linear model in each category is overlaid and compared to a grey dotted line representing a linear model produced without the breakdown into separate categories (i.e. 'ALL-ALL'). The linear model equation, correlation coefficient $r$, root mean squared error (RMSE) and number of pixels are included.

## 4.3 Spatial analysis of covariance between $SIF_{DS}$ and $GPP_{FX}$

The results of the analysis of covariance between pairs of vegetation covers within a climate zone, are shown in figure 6 through the $\eta^2$ for the slope and intercept of the linear relationship. It should be noted that the ANCOVA analysis assumes linearity between $SIF_{DS}$ and $GPP_{FX}$, which is present in most vegetation covers, with noted exceptions. Appendix A1 contains the full table of results, whilst similar analyses comparing the downscaled SIF - FluxSat GPP relationship and the TROPOMI SIF - FLUXCOM GPP relationship can be found in appendices A3 and A4 respectively.

The ANCOVA results in equatorial regions show that the categorisation by vegetation class is not a significant factor in the slope-dependence of the $SIF_{DS}$-$GPP_{FX}$ for all vegetation types except evergreen broadleaf forests, which, as discussed, exhibits non-linearity in the $SIF_{DS}$-$GPP_{FX}$ relationship. Differing intercepts between the DBF and the herbaceous vegetation covers, however, suggest that whilst the $SIF_{DS}$-$GPP_{FX}$ relationship scales in similar ways between vegetation covers, there may be differences in the starting potential. Linear relationships in grass and cropland are statistically indistinguishable, whilst around $10\%$ of the sum of squares between DBF and CRO/GRA intercepts can be attributed to the vegetation classification. In equatorial broadleaf forests $12-19\%$ of the difference in the $SIF_{DS}$-$GPP_{FX}$ scaling can be attributed to the categorisation, and therefore when using SIF as a proxy for productivity, EBF should clearly be considered separately from other vegetation classes.

In arid climates the difference between the slopes of vegetation covers is significant in terms of the p-value for all except the ENF-CRO pair. However, there is little to distinguish the $SIF_{DS}$-$GPP_{FX}$ scaling by vegetation categories, with less than $2\%$ of the sum of squares attributable to the vegetation covers for all except GRA-DBF ($7\%$). If the assumption is made that the vegetation categorisation has no effect on the $SIF_{DS}$-$GPP_{FX}$ slope, and that the slopes can be considered parallel between vegetation covers, then the intercepts generally distinguish between the woody and non-woody vegetation covers, with crossover in CRO-ENF. Between the ENF-DBF intercepts, $2\%$ of the sum of squares is attributable to the vegetation cover, whilst the proportion is $8\%$ for GRA-CRO. Mixing between herbaceous and woody covers, on the other hand, and the proportion of the sum of squares attributable to the vegetation cover is between $21-36\%$, with the exception of ENF-CRO, which are statistically almost indistinguishable.

In temperate regions the only major distinction in the gradient of the $SIF_{DS}$-$GPP_{FX}$ relationship between vegetation covers is found in deciduous broadleaf forests ($4-12\%$). As discussed in the previous section, temperate DBF is dominated by two distinct Northern and Southern hemisphere clusters with differing $SIF_{DS}$-$GPP_{FX}$ relationships, which results in a distinct and separate linear relationship. This feature is not observed in the TROPOMI SIF analysis. Regarding the other vegetation covers, assuming that the categorisation is of little importance to the slope, accounting for $\leq 2\%$ of the sum of squares, and that the slopes could be considered parallel between the vegetation covers, the differences in the intercept broadly divide along the lines of woody and herbaceous species. The sum of squares attributable to differences in the intercept are: woody-woody, $9\%$; herbaceous-herbaceous, $13\%$; woody-herbaceous, 27-68%.

Finally, in continental climates, ENF and DNF species exhibit a similar ($< 1\%$) $SIF_{DS}$-$GPP_{FX}$ scaling (though a much larger difference attributable to the intercept $30\%$) and are somewhat distinct from the other vegetation species ($6-11\%$), with the

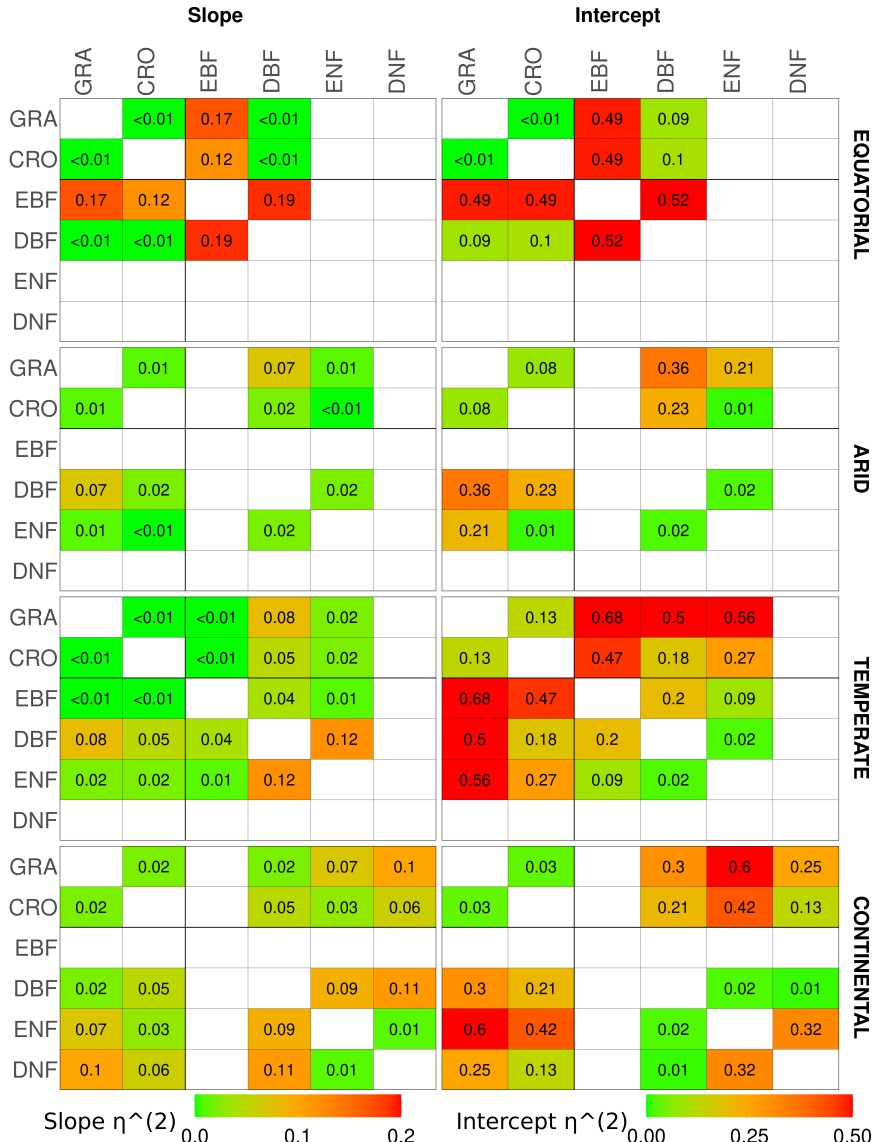

**Figure 6.** The $\eta^2$ parameter of an analysis of covariance between pairs of vegetation covers in different Köppen-Geiger climate groupings, for the slope (left) and intercept (right) of the linear relationship between downscaled SIF and FLUXCOM GPP. ANCOVA is performed on the intercept under the assumption that the difference between slopes is not significant. The $\eta^2$ parameter is comparable to the percentage of the difference in the slope or intercept (the latter assuming equivalence of the slopes) attributable to the difference in vegetation cover, with lower values signifying a smaller difference between vegetation covers. A slightly bolder line is used to separate the herbaceous species (CRO, GRA) from the woody species (EBF, DBF, ENF, DNF).

exception of CRO-ENF (3%). This feature in the slope of continental needleleaf forests is not observed in the analysis that uses FluxSat GPP in place of the FLUXCOM GPP. Within these other vegetation species, $< 5\%$ of the difference in the $\text{SIF}_{\text{DS}}$-$\text{GPP}_{\text{FX}}$ slope can be attributed to the choice of vegetation cover and, assuming the null hypothesis for the slope, the intercept again distinguishes between the herbaceous plants (GRA-CRO, 3%) and mixed, herbaceous-woody (DBF-CRO, 21%; DBF-GRA, 30%).

Overall, when analysing the the scaling of the $\text{SIF}_{\text{DS}}$-$\text{GPP}_{\text{FX}}$ response (i.e. the slope) between vegetation covers within a climate zone, the ANCOVA analysis suggests that there are large similarities, with potential slight exceptions in temperate deciduous broadleaf forests, continental needleaf forests, and the major exception of tropical evergreen forests. In terms of the scaling of the $\text{SIF}_{\text{DS}}$-$\text{GPP}_{\text{FX}}$ slope, these three vegetation covers may be treated as being reasonably distinct, with at least around 5% and up to 20% of the difference between slopes being attributable to the vegetation classification. Amongst the other species where the slope does not distinguish between vegetation covers so prominently (with generally less than 3% of the slope variation attributable to the vegetation categorisation), the intercept, and therefore the systemic difference between the linear relationships, loosely depends on whether the species is woody or herbaceous, with higher values for woody species. The difference in the $\text{SIF}_{\text{DS}}$-$\text{GPP}_{\text{FX}}$ response between cropland and grassland is particularly minor. A caveat must be made that there are some exceptions to these generalisations, and there is no statistically concrete global distinction between groupings of vegetation covers.

The results demonstrate that within a climate grouping there are broad similarities in the $\text{SIF}_{\text{DS}}$-$\text{GPP}_{\text{FX}}$ response of the considered vegetation classifications, excluding three key exceptions. When accounting for differences in the intercept, a loose possible grouping may be suggested of herbaceous and woody vegetation within each climate zones, with the exceptions of equatorial-EBF, temperate DBF, and continental forests (which can be fully distinguished when the difference in the intercept is considered, or split between broadleaf and needleleaf if considering only the scaling). This reduces the climate-vegetation categories for which we expect differing $\text{SIF}_{\text{DS}}$-$\text{GPP}_{\text{FX}}$ responses from 18 groups to 12 overall, with around three distinct groups in each climate zone, depending on the aggressiveness of the grouping.

## 4.4 Estimating the global spatial distribution of GPP with downscaled SIF

The mean growing season downscaled SIF can be projected into an estimate of growing season GPP using the global linear relationships for each climate and vegetation cover category displayed in figure 5. The absolute and percentage difference of this estimated GPP, $\text{GPP}_{\text{Est}}$, to the FLUXCOM GPP is shown in figure 7, which displays areas where the global, category-dependent, linear relationship shows positive or negative biases in replicating the $\text{GPP}_{\text{FX}}$ from the local $\text{SIF}_{\text{DS}}$ observations. The figures are created using three different versions of the global spatial linear relationships between $\text{SIF}_{\text{DS}}$ and $\text{GPP}_{\text{FX}}$. In the first instance, four separate linear relationships are derived for the four different climate zones. In the second instance the linear relationships are derived separately for each climate zone and vegetation cover, with 18 separate $\text{SIF}_{\text{DS}}$-$\text{GPP}_{\text{FX}}$ relationships. Finally, separate linear relationships are derived in each climate zone for each the different vegetation groupings suggested by the results of the analysis of covariance, with 12 groups overall.

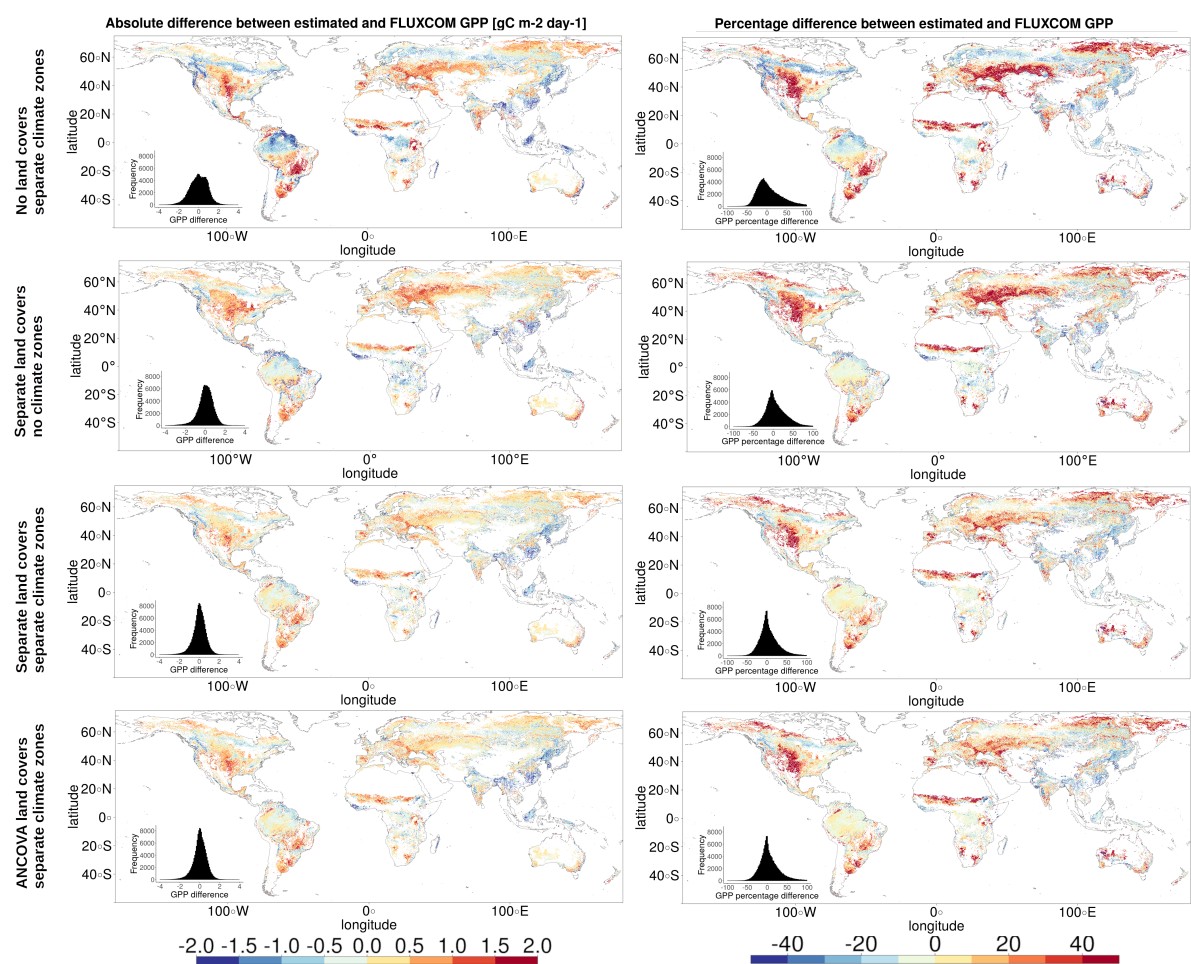

**Figure 7.** The absolute (left) and percentage (right) difference between the mean annual estimated primary production, $GPP_{Est}$ and the mean annual FLUXCOM GPP. $GPP_{Est}$ is estimated by projecting the downscaled SIF at each pixel using $SIF_{DS}$-$GPP_{FX}$ relationships derived within: top, each climate zone; upper middle, each vegetation cover; lower middle, each vegetation cover within each climate zone; bottom, different climate-vegetation groupings suggested by the analysis of covariance.

The figures show that there is added value for GPP prediction in breaking down the relationship into the differing vegetation covers since the $SIF_{DS}$-$GPP_{FX}$ relationship is not climate and vegetation invariant. When only the Köppen-Geiger climate grouping is used to classify the spatial $SIF_{DS}$-$GPP_{FX}$ relationships, there is a significantly greater difference between the FLUXCOM GPP and the GPP estimated from the downscaled SIF, compared to when vegetation cover is taken into account. As may be expected, the vegetation covers flagged as particularly distinguished in their spatio-temporal $SIF_{DS}$-$GPP_{FX}$ response,

such as equatorial evergreen forests and continental needleleaf forests, especially suffer from this lack of a breakdown. When only the vegetation covers are considered, and no climate grouping is proposed, there is a smaller difference between the estimated GPP and the FLUXCOM GPP than in the case of the climate groupings alone, suggesting that differences between

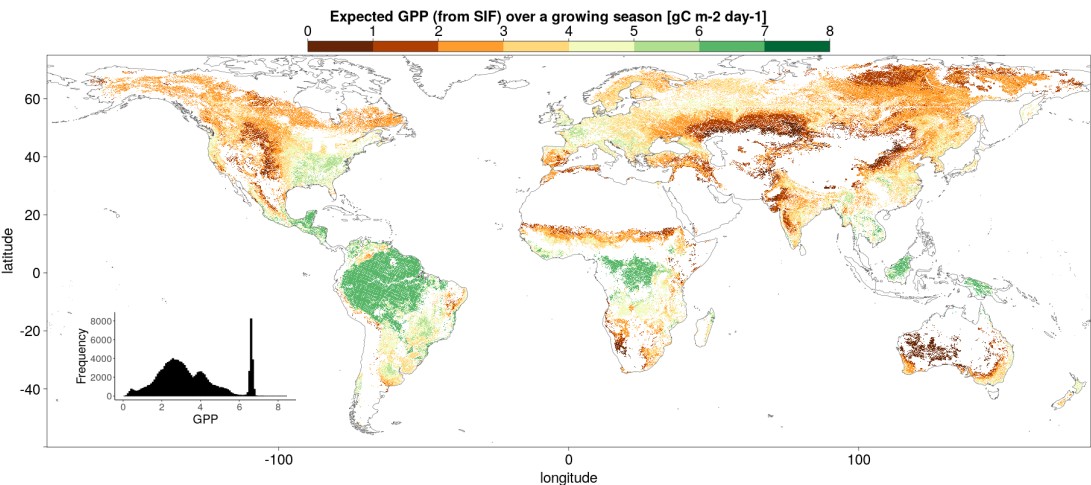

**Figure 8.** The global GPP estimated from the downscaled SIF and the $SIF_{DS}$-$GPP_{FX}$ spatial linear relationships between vegetation groupings suggested from the results of the analysis of covariance.

vegetation covers are more important in determining the $SIF_{DS}$-$GPP_{FX}$ relationship than the climate zone grouping. However there are still noticeable differences compared to the relationships that include a breakdown by climate grouping, as can be
seen in the width of the inset histograms. The similarity in the lower figures, where the $SIF_{DS}$-$GPP_{FX}$ scaling depends on the grouping suggested by the analysis of covariance, compared to the unique vegetation covers in the middle figures, show that whilst vegetation cover appears to be an important parameter in classifying $SIF_{DS}$-$GPP_{FX}$ relationships, it is possible to combine vegetation groups in a way that doesn't noticeably affect the $SIF_{DS}$-$GPP_{FX}$ scaling. It should be noted that vegetation cover here may be a proxy for other variables, such as local conditions, soil type or a refined climate grouping, and in this sense
further investigation in similar, localised conditions is required.

Figure 8 shows the global gross primary production estimated from the downscaled SIF and the $SIF_{DS}$-$GPP_{FX}$ relationships between vegetation groupings suggested by the ANCOVA results. It is particularly notable that in equatorial rainforests, the flat linear relationship derived between $SIF_{DS}$-$GPP_{FX}$ results in estimated GPP values with low variation.

### 4.5 The $SIF_{DS}$ response to meteorological fluctuations

Figure 9 shows the average z-score of the $SIF_{DS}$ with respect to the z-score of the four meteorological variables considered in this study as environmental drivers of primary productivity. Each bin contains multiple data points (at least five in order to be displayed) with the $SIF_{DS}$ z-score taken as the average of the data points within the bin. The figure shows how anomalies in monthly values of climate drivers - relative to the 'average' conditions for that month - are related to fluctuations in $SIF_{DS}$. The figures are broken down into the same categorisation as in the previous study, clearly showing that in the various climates,
vegetation covers respond differently, and sometimes in opposing directions, to climate drivers depending on the limiting

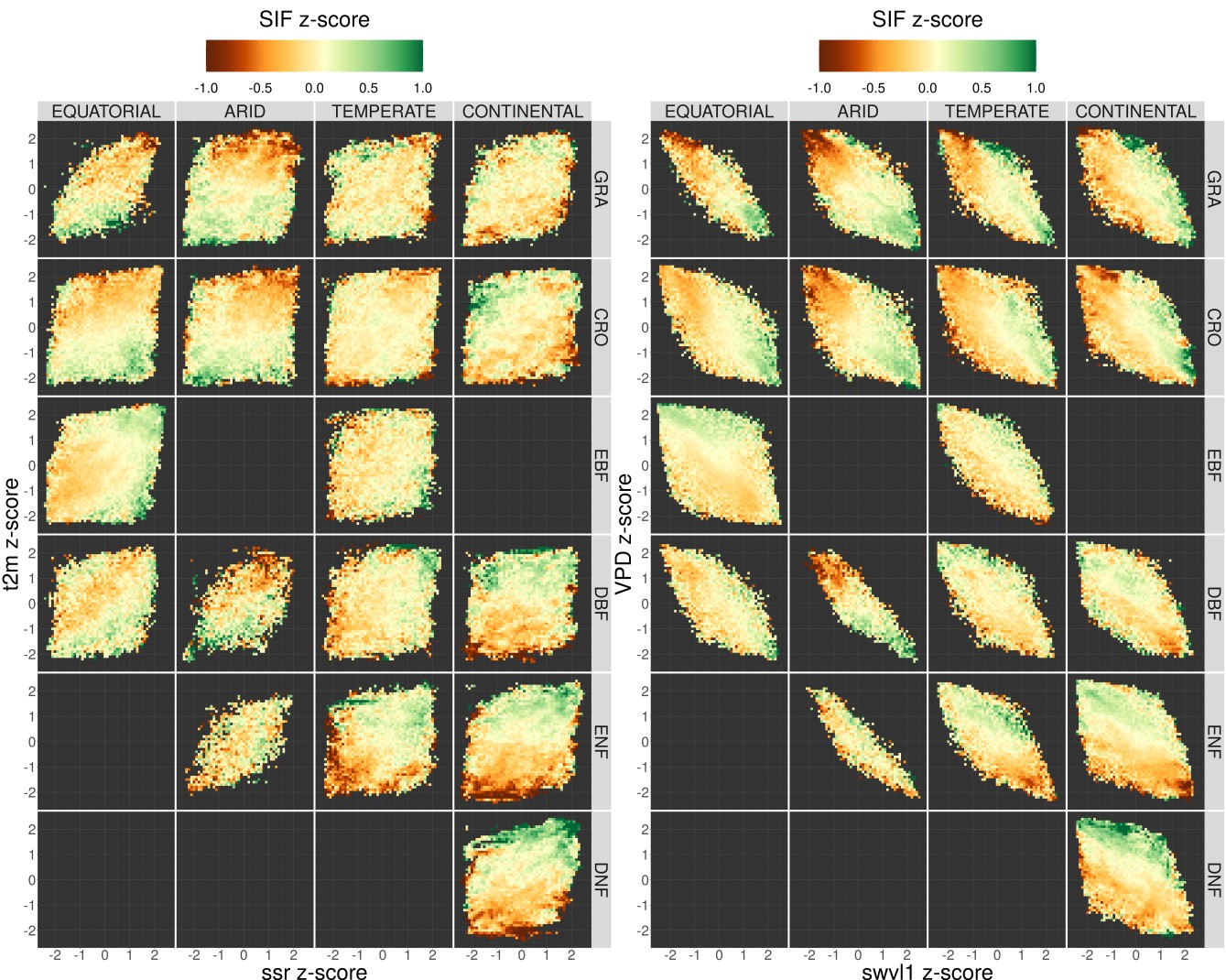

**Figure 9.** The relationship between fluctuations in meteorological variables and the corresponding fluctuations in the measured downscaled SIF. The fluctuations are measured relative to the monthly mean for each pixel and expressed as a z-score. The four meteorological variables are air temperature and net surface solar radiation (t2m and ssr, left) and vapour pressure deficit and soil moisture (VPD and swvl1, right). The SIF$_{DS}$ response in each bin is the mean of all data points in each bin of meteorological z-score. The data is broken down into separate Köppen-Geiger climate zones and land cover categories.

factor of photosynthesis (e.g. water scarcity, low temperatures, etc). Equivalent figures for the FLUXCOM GPP can be found in appendix A2.

Figure 10 shows the average SIF$_{DS}$ and GPP$_{FX}$ z-scores as a function of the corresponding z-score of the four meteorological variables. The figure uses the exact data that is input into figure 9, and categorises the temperature, VPD, soil moisture and

solar radiation z-score of pixels from the long-term monthly mean into 10 groups between -2.5 and +2.5. The median corresponding $SIF_{DS}$ and $GPP_{FX}$ z-scores, relative to the long-term monthly mean of each pixel, are shown for each meteorological variable. The results are broken down by the Köppen-Geiger climate and vegetation cover groupings discussed previously. The figure therefore shows the average $SIF_{DS}$ and $GPP_{FX}$ fluctuations that correspond to a given fluctuation in each meteorological condition, and can be used to interpret the meteorological drivers that may result in fluctuations in vegetation productivity.

The first point to note is that the link between $SIF_{DS}$ and meteorological fluctuations is more significant in some climate/vegetation cover categories than others. The $SIF_{DS}$ from grasslands and croplands responds in a very similar manner across all climates, but often differs from the response of woody vegetation. The $SIF_{DS}$ and $GPP_{FX}$ together respond in a similar way to the meteorological fluctuations, with the $GPP_{FX}$ generally more responsive, particularly in the case of woody vegetation. This is likely caused by the inclusion of meteorological information in the FLUXCOM GPP product, resulting in a correlation and

so over-sensitivity. Whilst the $SIF_{DS}$ may be less sensitive in general, unlike the FLUXCOM model it also captures information relating to the physiology of the plant, potentially bringing extra information into consideration when determining vegetation response.

    Clear and expected trends in the $SIF_{DS}$ data can be picked out. For example, plants in cooler climates respond more positively to higher temperature fluctuations and plants in arid climates benefit significantly from soil moisture and reduced VPD (more

humid conditions). Arid and continental climates, which in general are often harsher environments for plant life, exhibit a larger meteorological dependence than equatorial and temperate ones, whilst herbaceous plants are generally also more weather dependent. As the $SIF_{DS}$ response is measured with respect to conditions in an average month, the response often differs between Köppen-Geiger climates, for example, DBF and ENF forests respond positively to VPD (drier air) in temperate and continental climates but negatively in tropical and arid climates.

The response of $SIF_{DS}$ to fluctuations in the meteorological variables is not always of simple interpretation since there may be co-limitation from multiple drivers linked by complex correlation patterns. In figure 9, co-limitation is observed where the direction of the $SIF_{DS}$ response lies along the diagonal, for example the preference for both high temperatures and high levels of radiation in temperate deciduous broadleaf forests, or for conditions of low VPD and high soil moisture content in grassland and croplands. Co-dependence between the atmospheric variables means that it is difficult to directly explain

fluctuations in $SIF_{DS}$ via individual meteorological variables in isolation of the other meteorological variables, for example, the correlation between warmer temperatures and high VPD, results in a similar $SIF_{DS}$ response in cooler continental woody forests. Additionally, differences and sub-patterns in the $SIF_{DS}$-meteorological response may be complicated by the spatial distribution of plant species, which is not captured by the broad Köppen-Geiger categorisation. For example equatorial EBF forests may be located in wetter environments than Equatorial DBF forests, and therefore profit from differing fluctuations in

the local climate, in this case a lower soil moisture and higher VPD.

    Figure 10 shows that the strength of the relationship between the $SIF_{DS}$ fluctuations and the meteorological fluctuations generally increases for more extreme deviations of $SIF_{DS}$. For example, in continental deciduous needleleaf forests, a two standard deviation increase in the temperature (relative to the long-term mean for that month) corresponds to a $SIF_{DS}$ that is an average of 0.8 standard deviations higher than usual. In comparison, a smaller temperature fluctuation of between 1 and 1.5 standard de-

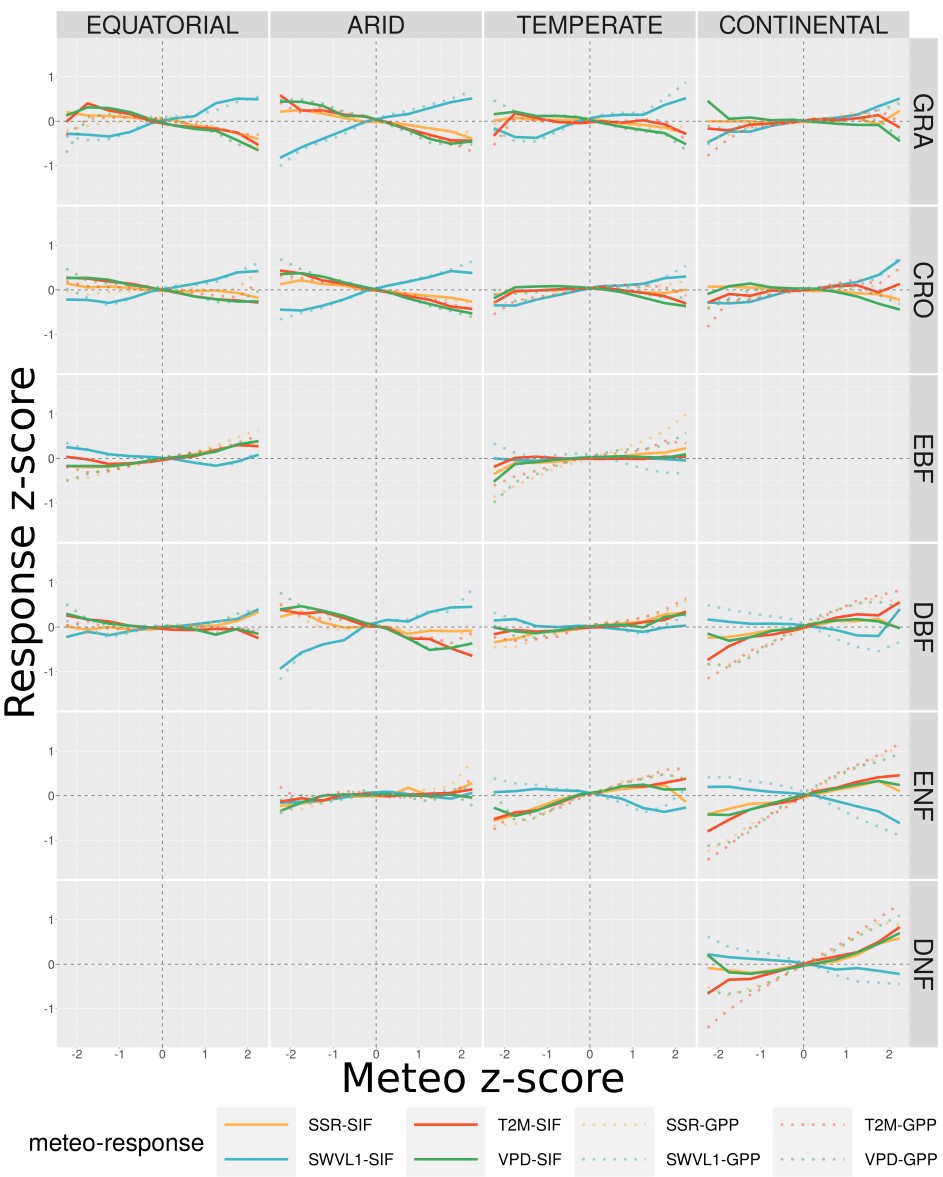

**Figure 10.** The average fluctuation in the remotely-sensed downscaled SIF and FLUXCOM GPP, as a function of the fluctuation in several meteorological variables. The fluctuations are defined in terms of the z-score of each variable, calculated for 10 different bins between -2.5 and 2.5, and the corresponding median $SIF_{DS}/GPP_{FX}$ z-score, relative to the long-term (2007-2014) monthly mean for each pixel (each pixel may contribute multiple months within the growing season). The meteorological variables considered from the Copernicus Climate Service (C3S) Climate Data Store (CDS) and are: surface net solar radiation (SSR), air temperature (T2M), vapour pressure deficit (VPD) and soil moisture (SWVL1). The data is broken down into separate Köppen-Geiger climate zones and land cover categories.

viations above the monthly mean temperature would correspond to a slightly lower increase in the SIF (+0.3 standard devations above the monthly mean). The results therefore provide evidence that not only do fluctuations in meteorological conditions correspond to fluctuations in SIF, but more extreme fluctuations often result in more extreme fluctuations in SIF. In this context the study suggests that it may be possible to use high-resolution SIF as a near-real time measure of the response of vegetation productivity to climate fluctuations, as well as demonstrating where vegetation may be resistant to certain fluctuations. For

example, evergreen broadleaf forests appear to show relatively little deviation in SIF up to reasonably extreme weather fluctuations. It is important to note though, that 'extreme fluctuations' here are measured relative to a location's average climate variation, which may be small in absolute terms compared to other categories. As the climate categorisation considered in this study is relatively broad, further research of using high resolution SIF on specific ecosystems is required.

Finally, the results are used to determine the driving and limiting climate variables on a global scale. Figure 11 shows a map
of the meteorological variable corresponding to the highest and lowest average SIF fluctuation.

## 5   Discussion

### 5.1   The use of downscaled SIF be used as a proxy for GPP: does it add value?

The study demonstrates the utility of the Duveiller et al. (2020) downscaling method in providing a robust, high-resolution SIF dataset that can be used as proxy for gross primary production. This method uses a light-use efficiency modelling based
approach to establish a relationship between SIF and higher resolution remote sensing variables. The resulting high resolution $SIF_{DS}$ benefits the analysis in enabling higher quality selections in the vegetation classification of pixels, and therefore more precision when assessing the different dynamics and patterns of the relationships between $SIF_{DS}$ and $GPP_{FX}$ across different vegetation covers and climate regions. A high level of spatio-temporal correlation is found across almost all climate and vegetation groupings, comparable to levels observed between non-downscaled SIF and FLUXCOM GPP. Breaking down the
correlations into their separate constituent vegetation covers shows diversity in the $SIF_{DS}$-$GPP_{FX}$ relationship, and therefore that there is some dependence on vegetation cover in the relationship between canopy-level SIF and vegetation productivity. For the most part, the downscaled SIF reproduces the spatial patterns observed in the FLUXCOM GPP data, for example in figure 2, and scales linearly, with a few notable exceptions.

The clear response of $SIF_{DS}$ to meteorological fluctuations of key climatic drivers shows that it is possible to observe the
temporal patterns and anomalies of vegetation productivity and stress remotely, via satellite. This suggests the possibility of using SIF in the near-real-time monitoring of vegetation reaction to environmental conditions. As climates change it becomes increasingly important to know how vegetation responds to both long-term trends in the climate as well as increasingly frequent extreme weather events.

The reproduction of known SIF-GPP patterns using the downscaled SIF demonstrates its utility as a high-resolution proxy
of primary productivity. In support of these conclusions, appendix A4 replicates the main analysis results with the substitution of a single year of TROPOMI data in place of the downscaled SIF, whilst appendix A3 ensures the conclusions are not unique to the choice of the GPP dataset. In this sense the analysis serves as a diagnostic benchmark for the comparison of SIF and GPP

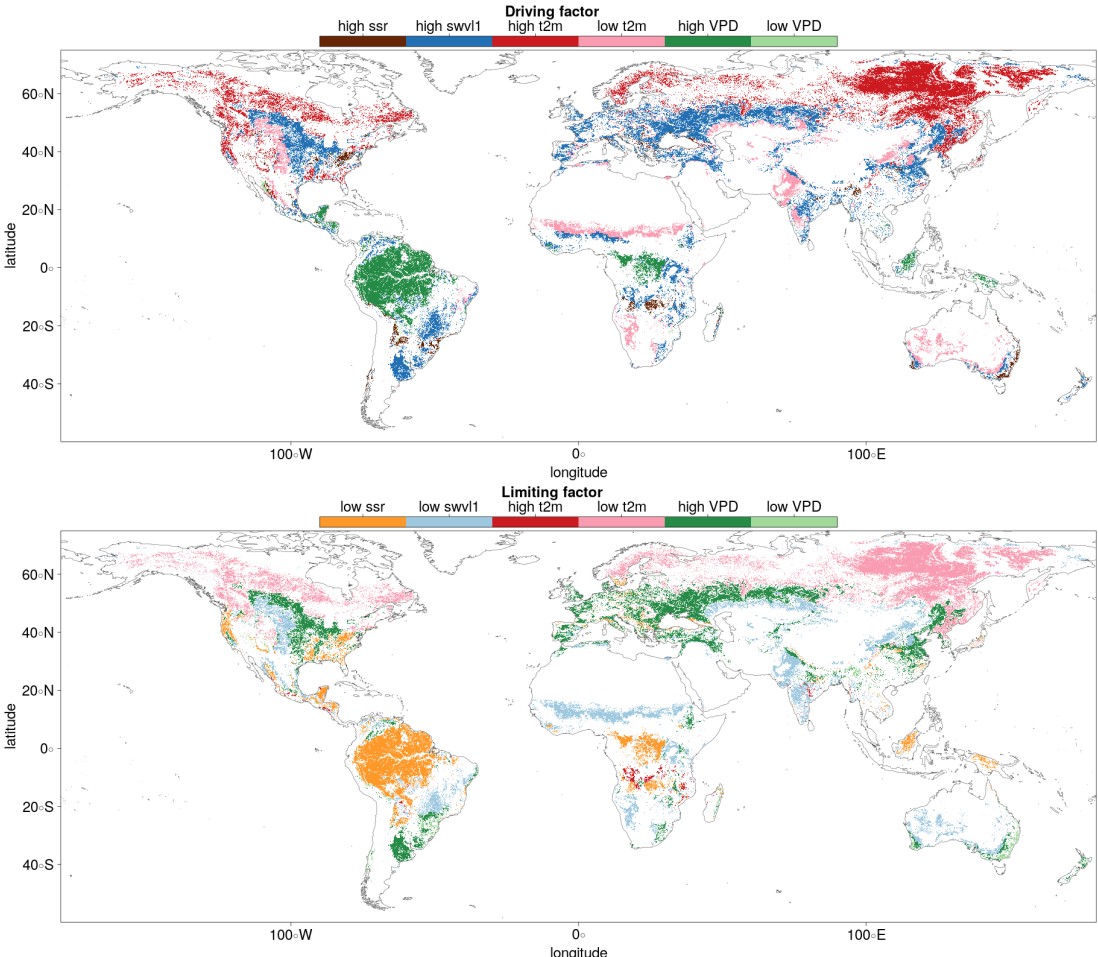

**Figure 11.** The driving (above) and limiting (below) meteorological conditions displayed at a globe scale. The driving and limiting variables are those that respectively correspond to the largest upward and downward fluctuations in downscaled SIF for each Köppen-Geiger climate-vegetation grouping.

datasets. The use of the downscaling method on recent and future retrievals of SIF, such as the high-resolution retrievals from the TROPOMI satellite instrument, will enable further study on the relationship between SIF and GPP. Furthermore, the current downscaled SIF dataset provides an archive at a comparable resolution for the analysis of trends across longer timescales.

## 5.2 Areas of divergence between SIF$_{DS}$ and GPP$_{FX}$

Figure 2 shows a clear divergence between the most productive areas in terms of FLUXCOM GPP, the equatorial rainforests of Brazil, central Africa and Indonesia, and the regions with the highest levels of downscaled SIF, the croplands of the mid-West, Western Europe and South America.

Equatorial broadleaf forests are also areas with reduced spatio-temporal correlation and scaling between $SIF_{DS}$ and $GPP_{FX}$, as seen in figure 3, with some areas anti-correlated. Figure 5 shows that the high variance in downscaled SIF is not matched by the similarly wide variation in FLUXCOM GPP observed in other vegetation types resulting in a flat relationship. This may hint at a saturation at high values of $GPP_{FX}$, whereby the observed $SIF_{DS}$ increases without a corresponding increase in $GPP_{FX}$ at the same rate. Such a plateau, particularly in evergreen broadleaf forests, could be driven by the saturation of the fraction of absorbed photosynthetically active radiation at high LAI values or perhaps by constraints in the $GPP_{FX}$ model. Indeed the
largest uncertainty in the FLUXCOM dataset is found in the tropics, an area with limited FLUXNET sites, and a similarly low correlation has been observed between $SIF_{DS}$ and $GPP_{FX}$ on a seasonal scale (Jung et al., 2020). A similar, if slightly reduced, plateau in the spatial $SIF_{DS}$-$GPP_{FX}$ relationship in evergreen needleleaf forests supports evidence of non-linearity in the temporal relationship found in Kim et al. (2021), similarly attributed to $GPP_{FX}$ saturation (as measured via an eddy covariance system) with absorbed photosynthetically active radiation.

The comparatively higher values of $SIF_{DS}$ in productive farm-belts supports evidence, such as in Guanter et al. (2014), that SIF-based crop productivity estimates are higher than other GPP estimates. The distribution of C3 and C4 crops may play a role here, as demonstrated by Zhang et al. (2017), which finds an underestimation of the FLUXCOM GPP in cropland areas, in addition to an overestimation in tropical rainforests. It may therefore be the case that the downscaled SIF is more sensitive
to C3/C4 differences than the FLUXCOM GPP model. Additionally, some studies, such as He et al. (2020) and Li and Xiao (2022), show more linearity between SIF and GPP in C4 crops, compared to C3 crops, with GPP estimated by eddy covariance towers. Differences in crop cover impacting the $SIF_{DS}$-$GPP_{FX}$ scaling may also be seen in figure 7. For example, in East Asia, there is an under-estimate of productivity based on the global $SIF_{DS}$-$GPP_{FX}$ relationship, and local measurements of downscaled SIF, whilst there is an over-estimate in the Americas, Africa and Europe.

The divergences between $SIF_{DS}$ and $GPP_{FX}$ may partially be attributed to the procedures used to collect and model the input data, however the divergences also support growing evidence of physiological reasons for the $SIF_{DS}$-$GPP_{FX}$ differences. This suggests that downscaled SIF could provide added value to the FLUXCOM estimate of the GPP in certain regions where the characterization of vegetation based on FaPAR and functional types in the machine learning framework is not sufficient to capture the spatio-temporal pattern of primary productivity.

## 5.3 The universality of the $SIF_{DS}$-$GPP_{FX}$ relationship across vegetation covers

Differences in the spatio-temporal $SIF_{DS}$-$GPP_{FX}$ correlation and linear relationship suggests that there is some deviation, on average, between vegetation covers. However, there is also substantial variability within vegetation groupings, meaning that for all except the clearest outliers, it is not possible to statistically distinguish between vegetation categories based on these deviations alone. Equatorial evergreen broadleaf forests clearly stand out as an outlier, with a spatio-temporal $SIF_{DS}$-
$GPP_{FX}$ relationship that is divergent from the other vegetation types and should be treated separately when projecting estimates of productivity from SIF, until the reasons for this divergence are fully accounted for. For the other vegetation covers with $SIF_{DS}$-$GPP_{FX}$ relationships that scale more linearly, there is no fixed $\eta^2$ threshold to categorically dictate when the vegetation categorisation plays an important role in distinguishing between the $SIF_{DS}$-$GPP_{FX}$ relationships. This is particularly true in

cases where the difference in the slope is small ($\eta$) but significant (p-value), whilst the difference in the intercept is large. Additionally, the intercept in the linear relationship tends to be slightly higher for woody trees compared to the herbaceous species and therefore a categorisation could loosely divide along this broad physiological plant trait.

The universality of the SIF-GPP relationship with respect to vegetation groupings is in area of active debate (Turner et al., 2021; Li and Xiao, 2022). Differences between vegetation covers likely result from differences is the canopy architecture and physiology, in particular the leaf clustering, chlorophyll content and maximum carboxylation capacity (Verrelst et al., 2015). This is particularly true for differences between herbaceous and woody vegetation, where for the latter, the lower photon escape probability from the canopy results in a lower intensity of SIF for a given productivity. Additionally, as discussed previously, further disaggregation of vegetation covers may be beneficial, for example distinguishing between deciduous broadleaf forests in Northern and Southern hemispheres, and between C3/C4 vegetation. Indeed it may be the case that there are more differences within certain vegetation covers, than between vegetation covers, and this effect may depend on the scale of the analysis. It is important to note however, that vegetation cover in the analysis may partially be a proxy for other factors or regional variables, such as background climate conditions and soil properties (Reichstein et al., 2014).

Distinctions between vegetation covers in the $SIF_{DS}$ response to meteorological fluctuations shows the divide is broadly along these lines of woody vs non-woody vegetation types. Herbaceous plants are more susceptible to changes in water supply than woody species, universally preferring high soil moisture and low vapour pressure deficit in all environments. For woody trees, vapour pressure deficit tends to be more important than soil moisture, and plays very little role at all in the $SIF_{DS}$ response of tropical evergreen broadleaf forests. This highlights the importance of using soil moisture, in addition to VPD, in quantifying droughts, and in particular its impacts on herbaceous vegetation (Stocker et al., 2018). Additionally, herbaceous species tend to respond stronger to meteorological fluctuations, with the exception of needleleaf forests. Overall, the study shows that it is possible to draw a distinction in the $SIF_{DS}$-$GPP_{FX}$ and $SIF_{DS}$-meteorological relationships between vegetation covers. These loosely divide between woody and herbaceous vegetation, with particular cases where further investigation is needed to fully understand the relationship dynamics.

## 6 Conclusions

This exploratory analysis confronts two observation-based products that inform on the spatio-temporal variability of primary productivity at global scale, highlighting areas of coherence and divergence. Firstly, it demonstrates the utility of the Duveiller et al. (2020) downscaling method in providing a high-resolution SIF dataset that can be used as proxy for gross primary production for specific vegetation covers. Secondly, in highlighting areas of divergence, the study provides a remotely-sensed, independent comparison of downscaled SIF with the FLUXCOM GPP model. The relatively fine resolution of the downscaled SIF enables a global exploration of the spatio-temporal relationship between $SIF_{DS}$ and $GPP_{FX}$ at a level that distinguishes between differing vegetation cover types, enabling a categorisation of vegetation covers based on the $SIF_{DS}$-$GPP_{FX}$ response. For the most part, the gradient of the spatial $SIF_{DS}$-$GPP_{FX}$ response is similar between differing vegetation types, with the exception of equatorial broadleaf forests, and, potentially, slight exceptions in continental needleleaf forests and temperate

deciduous broadleaf forests. However, the $GPP_{FX}$ systematic potential for a given $SIF_{DS}$ observation displays more variation between species, with some divergence between woody and non-woody plants. The study provides both evidence for the spatio-temporal correlation between downscaled SIF and FLUXCOM GPP, with different climate and vegetation covers exhibiting variability in the $SIF_{DS}$-$GPP_{FX}$ relationship. The temporal component of the $SIF_{DS}$-$GPP_{FX}$ relationship is generally stronger than the spatial component, in particular at an intra-annual scale. Regions of climate and vegetation cover exhibiting high spatial correlation between $SIF_{DS}$ and $GPP_{FX}$ also tend to exhibit higher temporal correlation, suggesting that the mechanisms driving spatial and temporal variability are similar. Vegetation in some climates, such as tropical rainforests, shows divergence from linearity in the $SIF_{DS}$-$GPP_{FX}$ relationship. Here the downscaled SIF data may provide additional, independent information to the FLUXCOM model, particularly at high $GPP_{FX}$ values where the model may be at risk of saturation of photosynthetically active radiation.

The study also demonstrates the possibility of using near real-time satellite SIF measurements to study the response of vegetation to meteorological anomalies over short (monthly) timescales. Proving this technique at a global scale demonstrates that high-resolution SIF responds to meteorological fluctuations in a similar way to FLUXCOM GPP. As such it has potential as a near real-time indicator of vegetation status that, unlike FLUXCOM GPP, is independent of meteorological variables on aggregate. Whilst there is similarity in the $SIF_{DS}$-$GPP_{FX}$ response between vegetation covers, there is more diversity between different vegetation covers in the $SIF_{DS}$ response to meteorological fluctuations, particularly between herbaceous species and woody trees.

The further collection of high-resolution SIF data via the downscaling method of Duveiller et al. (2020) in addition to satellites such as OCO-2 and the future FLEX mission, will continue aid the understanding of the relationship between SIF, environmental conditions and plant productivity, as well as the variety of response between vegetation covers. The benefit will be to advance our understanding and estimation of the Earth's productivity, on both a local and global level.

*Data availability.* The following public datasets are used in the analysis:

– **Vegetation cover data** is provided by the Copernicus Climate Change Service (C3S) via the climate data store platform, with the data created by the ESA CCI program (CCI, 2017; Defourny, 2019),
https://cds.climate.copernicus.eu/cdsapp#!/dataset/satellite-land-cover

– **Climate zone classification** follows the Köppen-Geiger climate classification scheme (Kottek et al., 2006),
http://koeppen-geiger.vu-wien.ac.at/present.htm

– **Growing seasons** are defined by the Vegetation Index and Phenology (VIP) global dataset from NASA's Making Earth System Data Records for Use in Research Environments (MEaSUREs) program (Didan, 2016),
https://lpdaac.usgs.gov/products/vipphen_ndviv004/

– **Downscaled SIF** data is provided by Duveiller et al. (2019)
https://data.jrc.ec.europa.eu/dataset/21935ffc-b797-4bee-94da-8fec85b3f9e1

  – **Gross primary productivity** data is provided by the FLUXCOM project (Jung and FLUXCOM Team, 2016), https://www.bgc-jena.mpg.de/geodb/projects/FileDetails.php

  – **Meteorological data** is obtained from the ERA5-Land monthly reanalysis dataset (Muñoz Sabater, 2019b), https://cds.climate.copernicus.eu/cdsapp#!/dataset/reanalysis-era5-land

  – **TROPOMI SIF data** is provided by the ESA –TROPOSIF project (Guanter et al., 2021), https://eo4society.esa.int/projects/sentinel-5p-innovation-solar-induced-chlorophyll-fluorescence-sif/

  – **FluxSat GPP data** is obtained from ORNL Distributed Active Archive Center (Joiner and Yoshida, 2021) https://daac.ornl.gov/cgi-bin/dsviewer.pl?ds_id=1835

## Appendix A

### A1    ANCOVA results for downscaled SIF and FLUXCOM GPP

Table A1 contains the full results of the analysis of covariance for the $SIF_{DS}$-$GPP_{FX}$ relationship between pairs of land covers. In each climate category, vegetation cover pairs with the largest $\eta^2$ for the slope are listed first, where the slope is significant (p-value < 0.05). If differences in the regression slope are not significant (i.e. the slopes are considered to be parallel), then the difference in the size of the effect of the intercept is considered, such that the lowest ranked pairs within a climate zone are the most similar in their $SIF_{DS}$-$GPP_{FX}$ response.

### A2    FLUXCOM GPP response to meteorological fluctuations

Figure A1 shows the relationship between fluctuations in meteorological variables and the corresponding fluctuations in the FLUXCOM GPP.

| | land cover | | slope | | intercept | |
|---|---|---|---|---|---|---|
| Climate | $LC_1$ | $LC_2$ | p-value | $\eta^2$ | p-value | $\eta^2$ |
| Equatorial | EBF | DBF | $1.34 \times 10^{-95}$ | 0.19 | $< 1.00 \times 10^{-99}$ | 0.53 |
| Equatorial | EBF | GRA | $1.59 \times 10^{-77}$ | 0.17 | $< 1.00 \times 10^{-99}$ | 0.49 |
| Equatorial | EBF | CRO | $1.85 \times 10^{-55}$ | 0.12 | $< 1.00 \times 10^{-99}$ | 0.49 |
| Equatorial | DBF | CRO | $1.53 \times 10^{-01}$ | $< 0.01$ | $3.91 \times 10^{-45}$ | 0.10 |
| Equatorial | DBF | GRA | $1.36 \times 10^{-01}$ | $< 0.01$ | $1.37 \times 10^{-39}$ | 0.09 |
| Equatorial | GRA | CRO | $8.38 \times 10^{-01}$ | $< 0.01$ | $5.09 \times 10^{-01}$ | $< 0.01$ |
| Arid | GRA | DBF | $4.35 \times 10^{-28}$ | 0.07 | $< 1.00 \times 10^{-99}$ | 0.36 |
| Arid | CRO | DBF | $3.01 \times 10^{-10}$ | 0.02 | $6.88 \times 10^{-91}$ | 0.23 |
| Arid | DBF | ENF | $4.64 \times 10^{-06}$ | 0.02 | $1.75 \times 10^{-05}$ | 0.02 |
| Arid | GRA | CRO | $1.41 \times 10^{-04}$ | 0.01 | $3.14 \times 10^{-36}$ | 0.08 |
| Arid | GRA | ENF | $1.00 \times 10^{-03}$ | 0.01 | $6.46 \times 10^{-79}$ | 0.21 |
| Arid | ENF | CRO | $6.36 \times 10^{-01}$ | $< 0.01$ | $4.27 \times 10^{-05}$ | $< 0.01$ |
| Temperate | DBF | ENF | $4.09 \times 10^{-56}$ | 0.12 | $3.08 \times 10^{-08}$ | 0.02 |
| Temperate | DBF | GRA | $5.80 \times 10^{-39}$ | 0.08 | $< 1.00 \times 10^{-99}$ | 0.50 |
| Temperate | DBF | CRO | $2.36 \times 10^{-23}$ | 0.05 | $5.81 \times 10^{-86}$ | 0.18 |
| Temperate | EBF | DBF | $1.80 \times 10^{-19}$ | 0.04 | $< 1.00 \times 10^{-99}$ | 0.20 |
| Temperate | ENF | GRA | $1.77 \times 10^{-12}$ | 0.02 | $1.00 \times 10^{-99}$ | 0.56 |
| Temperate | ENF | CRO | $6.22 \times 10^{-08}$ | 0.02 | $< 1.00 \times 10^{-99}$ | 0.27 |
| Temperate | EBF | ENF | $2.33 \times 10^{-07}$ | 0.01 | $2.25 \times 10^{-42}$ | 0.09 |
| Temperate | EBF | GRA | $9.52 \times 10^{-01}$ | $< 0.01$ | $1.00 \times 10^{-99}$ | 0.68 |
| Temperate | EBF | CRO | $9.14 \times 10^{-01}$ | $< 0.01$ | $< 1.00 \times 10^{-99}$ | 0.47 |
| Temperate | GRA | CRO | $8.28 \times 10^{-01}$ | $< 0.01$ | $1.55 \times 10^{-63}$ | 0.13 |
| Continental | DBF | DNF | $5.72 \times 10^{-54}$ | 0.11 | $1.71 \times 10^{-07}$ | 0.01 |
| Continental | DNF | GRA | $3.77 \times 10^{-49}$ | 0.10 | $< 1.00 \times 10^{-99}$ | 0.25 |
| Continental | DBF | ENF | $4.57 \times 10^{-43}$ | 0.09 | $9.59 \times 10^{-11}$ | 0.02 |
| Continental | ENF | GRA | $1.27 \times 10^{-34}$ | 0.07 | $< 1.00 \times 10^{-99}$ | 0.60 |
| Continental | DNF | CRO | $1.54 \times 10^{-26}$ | 0.06 | $9.95 \times 10^{-62}$ | 0.13 |
| Continental | DBF | CRO | $9.06 \times 10^{-23}$ | 0.05 | $< 1.00 \times 10^{-99}$ | 0.21 |
| Continental | ENF | CRO | $8.43 \times 10^{-15}$ | 0.03 | $< 1.00 \times 10^{-99}$ | 0.42 |
| Continental | DBF | GRA | $1.66 \times 10^{-11}$ | 0.02 | $< 1.00 \times 10^{-99}$ | 0.30 |
| Continental | GRA | CRO | $2.66 \times 10^{-09}$ | 0.02 | $4.24 \times 10^{-14}$ | 0.03 |
| Continental | ENF | DNF | $9.18 \times 10^{-05}$ | 0.01 | $< 1.00 \times 10^{-99}$ | 0.32 |

**Table A1.** Analysis of covariance between pairs of land covers in different Köppen-Geiger climate groupings. ANCOVA is only performed on the intercept when the difference between slopes is not considered significant.

## A3  Comparison of downscaled SIF with FluxSat GPP

### A3.1  FluxSat GPP data

The presented analysis is repeated with an alternative GPP product to verify that the conclusions drawn regarding the nature of the spatial SIF-GPP relationship are not unique to the dataset. FluxSat is a global $0.05°$ GPP estimate derived from the MODIS Nadir Bidirectional Reflectance Distribution Function (BRDF)-Adjusted Reflectances, input into neural networks that upscale GPP estimated from FLUXNET eddy covariance tower sites (Joiner and Yoshida, 2021).

The data is aggregated to 8-day time steps and the growing season GPP is averaged over the period 2007-2014 in order to ensure compatibility with the downscaled SIF data. The pixels considered in the analysis are the same as those used in the main paper, and the methodology used is the same.

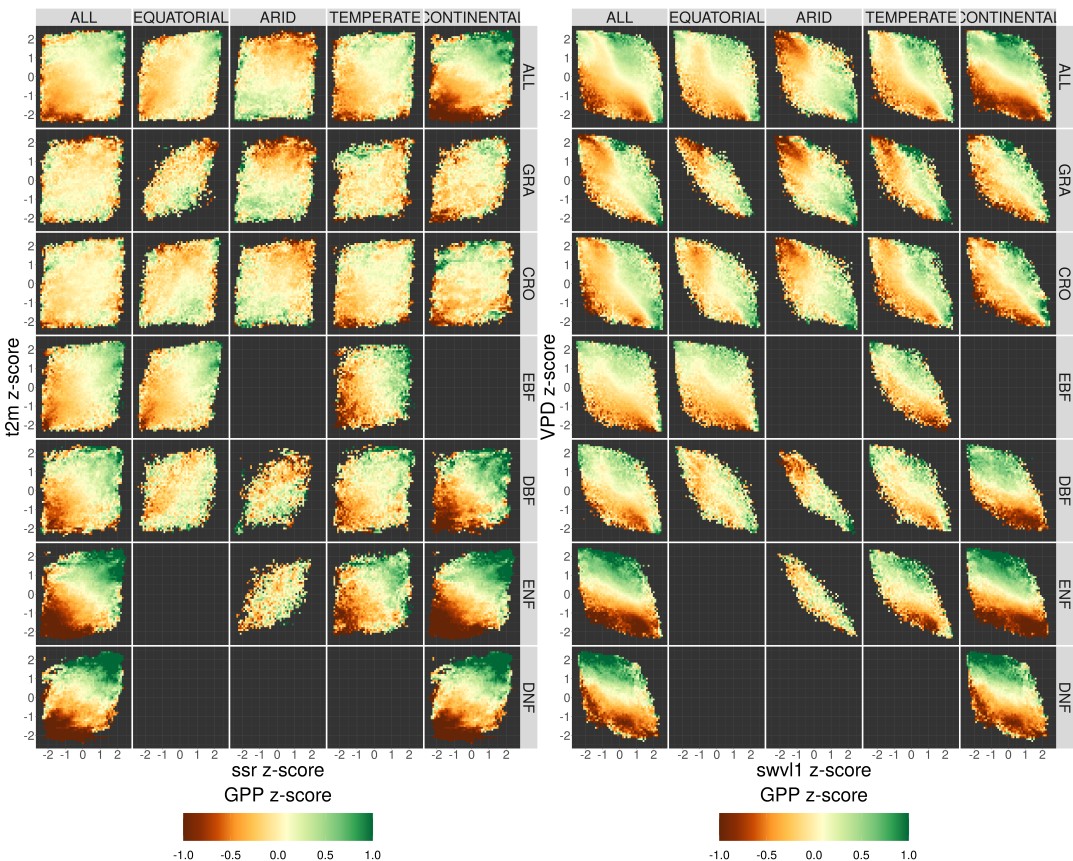

**Figure A1.** The relationship between fluctuations in meteorological variables and the corresponding fluctuations in the FLUXCOM GPP. The fluctuations are measured relative to the monthly mean for each pixel and expressed as a z-score. The four meteorological variables are air temperature and net surface solar radiation (t2m and ssr, left) and vapour pressure deficit and soil moisture (VPD and swvl1, right). The GPP$_{FX}$ response in each bin is the mean of all data points in each bin of meteorological z-score. The data is broken down into separate Köppen-Geiger climate zones and land cover categories.

### A3.2   FluxSat GPP distribution

Figure A2 shows the spatial distribution of the mean growing season FluxSat GPP and the difference to the mean growing season FLUXCOM GPP. The figure is comparable to that of downscaled SIF distribution, figure 2. The figures shows that there is a significant difference between FluxSat and FLUXCOM GPP across most of the world.

### A3.3   Spatial relationship between downscaled SIF and FluxSat GPP

Figure A3 shows the relative distribution and spatial linear relationship between the mean growing season FluxSat GPP as a function of the respective mean values of the downscaled SIF. The data are broken down into separate categories depending on

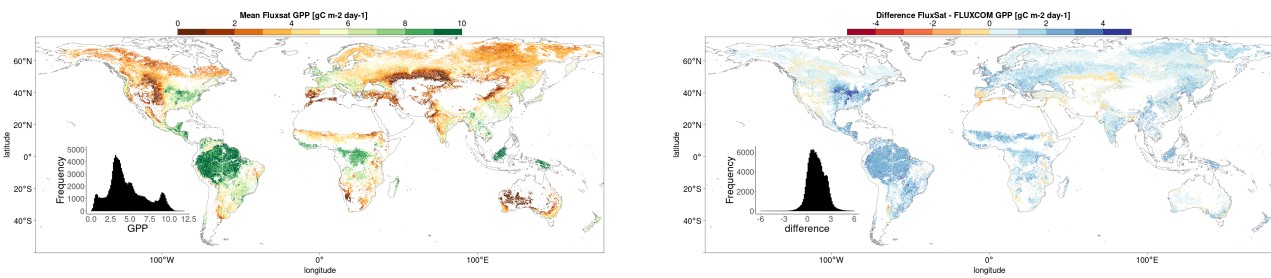

**Figure A2.** Left: The mean growing season FluxSat GPP 2007-2014. Right: The difference (FluxSat - FLUXCOM) between mean FluxSat GPP and mean FLUXCOM GPP (2007-2014).

the Köppen-Geiger climate grouping and dominant vegetation cover of the pixel. The figures show spatial correlations for the downscaled SIF and FluxSat GPP, are generally higher than those of the downscaled SIF and FLUXCOM GPP and so exhibit greater linearity.

### A3.4 Spatial analysis of covariance between downscaled SIF and FluxSat GPP

The ANCOVA analysis is repeated for the downscaled SIF with FluxSat GPP, and the $\eta^2$ parameters for slope and intercept are displayed in figure A4, whilst the full results can be found in table A2.

The results support the analysis of covariance between the downscaled SIF and the FLUXCOM GPP. The difference in the SIF-GPP scaling between vegetation covers is relatively unimportant with a the exceptions of equatorial evergreen broadleaf forests and temperate deciduous broadleaf forests. Continental needleleaf forests are less of an exception when the FluxSat GPP is considered however. Indeed, in general, the differences between vegetation covers are less prominent with the FluxSat GPP. There is less difference in the scaling of the SIF-GPP relationship between land covers, than there is in the starting potential, with the significance and the magnitude of effect of the choice of vegetation covers on the intercept slightly greater in comparisons between a woody and a non-woody species, than within woody/non-woody groupings. This intercept is generally higher for woody vegetation.

### A4 Comparison with TROPOMI SIF

#### A4.1 TROPOMI SIF data

As described in Duveiller et al. (2020) the downscaled SIF dataset is independently validated with OCO-2 SIF observations and, after bias correction, the resulting downscaled SIF data show high spatio-temporal agreement with the first SIF retrievals from the TROPOMI mission. Further comparison of the length-of-day corrected TROPOMI data with FLUXCOM GPP is provided to support the specific analysis presented in this paper (Guanter et al., 2021).

The TROPOMI data is averaged to 8-day time steps with the composite containing observations with a zenith angle below $40°$. The pixels considered in the analysis are the same as those used in the main paper, with the requirements on missing data

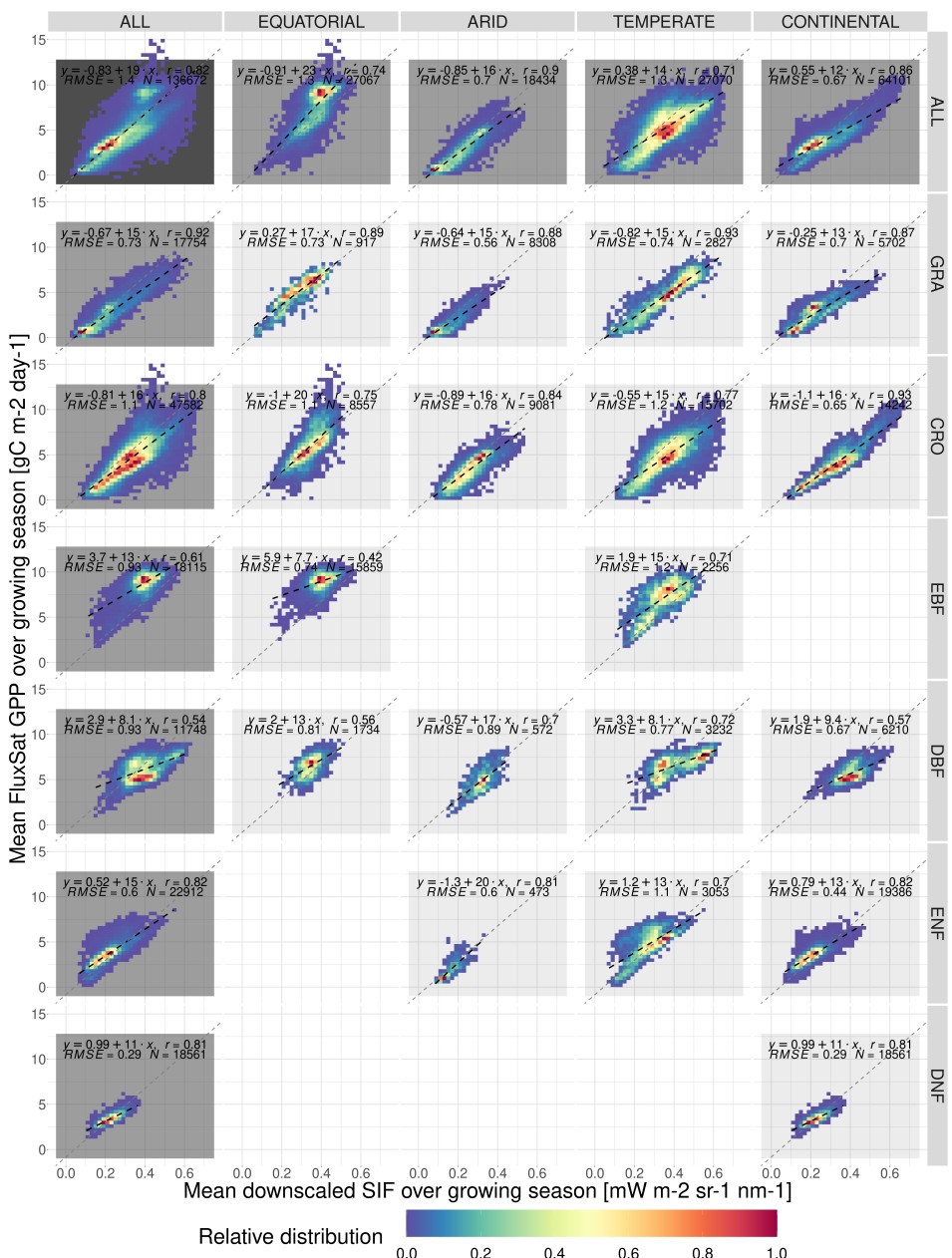

**Figure A3.** The spatial relationship between the mean growing season downscaled SIF and FluxSat GPP, broken down into separate Köppen-Geiger climate zones and vegetation cover categories. The plot shows the frequency distribution of pixels into $SIF_{DS}$-$GPP_{FX}$ bins, relative to the highest frequency bin in that category. A black dashed line representing a linear model in each category is overlaid and compared to a grey dotted line representing a linear model produced without the breakdown into separate categories (i.e. 'ALL-ALL'). The linear model equation, correlation coefficient $r$, root mean squared error (RMSE) and number of pixels are included.

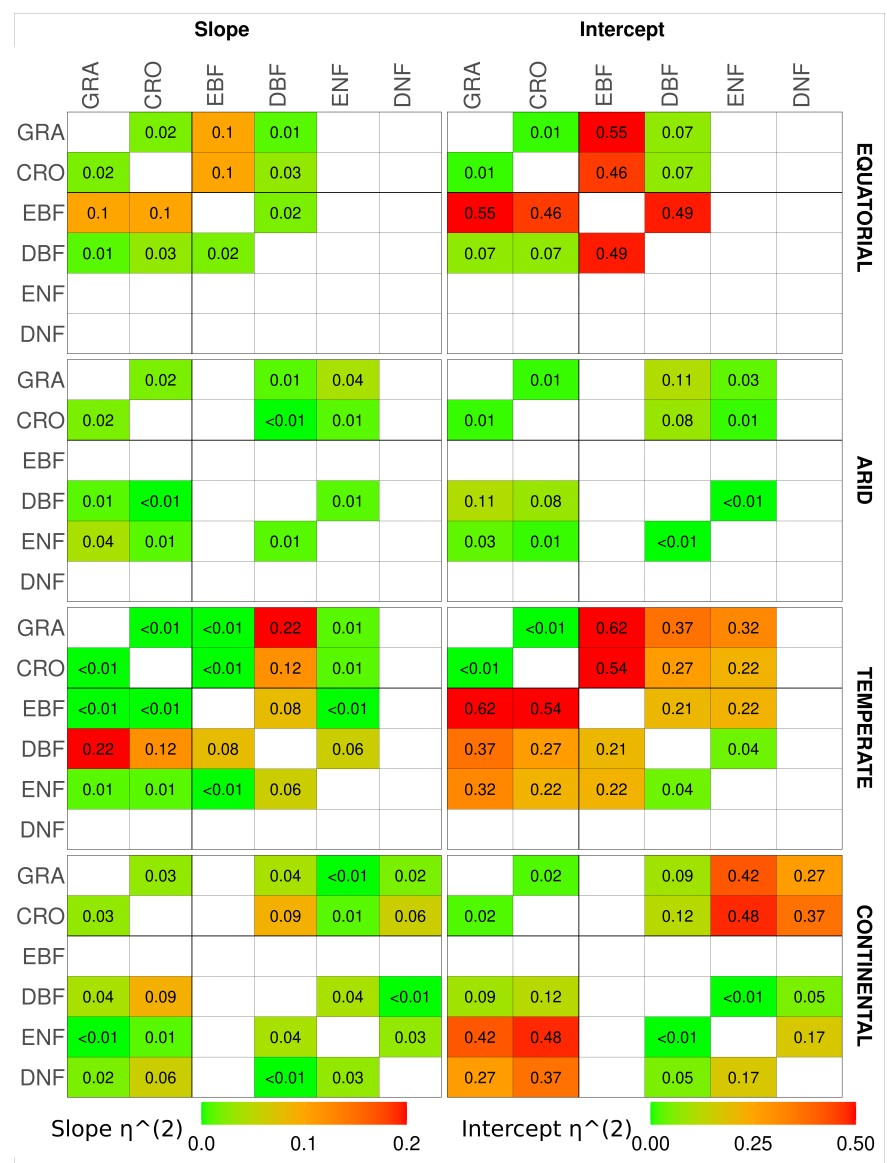

**Figure A4.** The $\eta^2$ parameter of an analysis of covariance between pairs of vegetation covers in different Köppen-Geiger climate groupings, for the slope (left) and intercept (right) of the linear relationship between downscaled SIF and FluxSat GPP. ANCOVA is performed on the intercept under the assumption that the difference between slopes is not significant. The $\eta^2$ parameter is comparable to the percentage of the difference in the slope or intercept (the latter assuming equivalence of the slopes) attributable to the difference in vegetation cover, with lower values signifying a smaller difference between vegetation covers. A slightly bolder line is used to separate the herbaceous species (CRO, GRA) from the woody species (EBF, DBF, ENF, DNF).

| Climate | land cover | | slope | | intercept | |
|---|---|---|---|---|---|---|
| | $LC_1$ | $LC_2$ | p-value | $\eta^2$ | p-value | $\eta^2$ |
| EQUATORIAL | EBF | DBF | $6.24\text{x}10^{-11}$ | 0.02 | $< 1.00\text{x}10^{-99}$ | 0.49 |
| EQUATORIAL | EBF | GRA | $1.82\text{x}10^{-46}$ | 0.10 | $< 1.00\text{x}10^{-99}$ | 0.55 |
| EQUATORIAL | EBF | CRO | $2.82\text{x}10^{-48}$ | 0.10 | $< 1.00\text{x}10^{-99}$ | 0.46 |
| EQUATORIAL | DBF | CRO | $3.50\text{x}10^{-15}$ | 0.03 | $2.15\text{x}10^{-31}$ | 0.07 |
| EQUATORIAL | DBF | GRA | $1.43\text{x}10^{-07}$ | 0.01 | $1.23\text{x}10^{-31}$ | 0.07 |
| EQUATORIAL | GRA | CRO | $4.85\text{x}10^{-09}$ | 0.02 | $5.15\text{x}10^{-05}$ | 0.01 |
| ARID | GRA | DBF | $6.80\text{x}10^{-06}$ | 0.01 | $5.25\text{x}10^{-42}$ | 0.11 |
| ARID | CRO | DBF | $4.02\text{x}10^{-01}$ | $< 0.01$ | $2.19\text{x}10^{-30}$ | 0.08 |
| ARID | DBF | ENF | $1.40\text{x}10^{-02}$ | 0.01 | $1.25\text{x}10^{-01}$ | $< 0.01$ |
| ARID | GRA | CRO | $2.14\text{x}10^{-08}$ | 0.02 | $2.84\text{x}10^{-05}$ | 0.01 |
| ARID | GRA | ENF | $3.98\text{x}10^{-16}$ | 0.04 | $5.99\text{x}10^{-11}$ | 0.03 |
| ARID | ENF | CRO | $8.20\text{x}10^{-05}$ | 0.01 | $2.61\text{x}10^{-05}$ | 0.01 |
| TEMPERATE | DBF | ENF | $4.00\text{x}10^{-27}$ | 0.06 | $3.94\text{x}10^{-17}$ | 0.04 |
| TEMPERATE | DBF | GRA | $< 1.00\text{x}10^{-99}$ | 0.22 | $< 1.00\text{x}10^{-99}$ | 0.37 |
| TEMPERATE | DBF | CRO | $4.62\text{x}10^{-57}$ | 0.12 | $< 1.00\text{x}10^{-99}$ | 0.27 |
| TEMPERATE | EBF | DBF | $6.07\text{x}10^{-40}$ | 0.08 | $< 1.00\text{x}10^{-99}$ | 0.21 |
| TEMPERATE | ENF | GRA | $1.59\text{x}10^{-06}$ | 0.01 | $< 1.00\text{x}10^{-99}$ | 0.32 |
| TEMPERATE | ENF | CRO | $3.47\text{x}10^{-04}$ | 0.01 | $< 1.00\text{x}10^{-99}$ | 0.22 |
| TEMPERATE | EBF | ENF | $9.00\text{x}10^{-03}$ | $< 0.01$ | $< 1.00\text{x}10^{-99}$ | 0.22 |
| TEMPERATE | EBF | GRA | $3.42\text{x}10^{-01}$ | $< 0.01$ | $< 1.00\text{x}10^{-99}$ | 0.62 |
| TEMPERATE | EBF | CRO | $5.23\text{x}10^{-01}$ | $< 0.01$ | $< 1.00\text{x}10^{-99}$ | 0.54 |
| TEMPERATE | GRA | CRO | $8.66\text{x}10^{-01}$ | $< 0.01$ | $9.95\text{x}10^{-01}$ | $< 0.01$ |
| CONTINENTAL | DBF | DNF | $3.00\text{x}10^{-03}$ | $< 0.01$ | $1.21\text{x}10^{-24}$ | 0.05 |
| CONTINENTAL | DNF | GRA | $1.92\text{x}10^{-08}$ | 0.02 | $< 1.00\text{x}10^{-99}$ | 0.27 |
| CONTINENTAL | DBF | ENF | $2.96\text{x}10^{-20}$ | 0.04 | $6.30\text{x}10^{-02}$ | $< 0.01$ |
| CONTINENTAL | ENF | GRA | $1.79\text{x}10^{-01}$ | $< 0.01$ | $< 1.00\text{x}10^{-99}$ | 0.42 |
| CONTINENTAL | DNF | CRO | $4.35\text{x}10^{-29}$ | 0.06 | $< 1.00\text{x}10^{-99}$ | 0.37 |
| CONTINENTAL | DBF | CRO | $4.96\text{x}10^{-43}$ | 0.09 | $1.81\text{x}10^{-55}$ | 0.12 |
| CONTINENTAL | ENF | CRO | $8.07\text{x}10^{-06}$ | 0.01 | $< 1.00\text{x}10^{-99}$ | 0.48 |
| CONTINENTAL | DBF | GRA | $2.11\text{x}10^{-17}$ | 0.04 | $2.72\text{x}10^{-44}$ | 0.09 |
| CONTINENTAL | GRA | CRO | $1.07\text{x}10^{-13}$ | 0.03 | $2.22\text{x}10^{-12}$ | 0.02 |
| CONTINENTAL | ENF | DNF | $5.12\text{x}10^{-16}$ | 0.03 | $3.44\text{x}10^{-81}$ | 0.17 |

**Table A2.** Analysis of covariance between pairs of land covers in different Köppen-Geiger climate groupings for the relationship between downscaled SIF and FluxSat GPP. ANCOVA is only performed on the intercept when the difference between slopes is not considered significant.

points loosened to ensure coverage. Due to the shorter time span of available data from TROPOMI, only the 2020 dataset is analysed here. Additionally, the analysis differs from that presented in the paper as the coverage of the VIPPHEN phenology dataset does not extend to the years covered by TROPOMI and the growing season of 2014 - the final VIPPHEN year available - is used to define the growing season and compare SIF and GPP. Finally the comparison is made with an extended FLUXCOM GPP dataset which may contain methodological differences from that used in the paper.

## A4.2  TROPOMI SIF distribution

Figure A5 shows the spatial distribution of the mean growing season TROPOMI SIF and the difference to the mean growing season downscaled SIF. The figure is comparable to that of downscaled SIF distribution, figure 2. TROPOMI generally shows a lower SIF than the downscaled values, with a mean difference of -0.077. These are relatively evenly distributed across the globe, with the exception of the tropics, which shows an excess in TROPOMI compared to downscaled SIF.

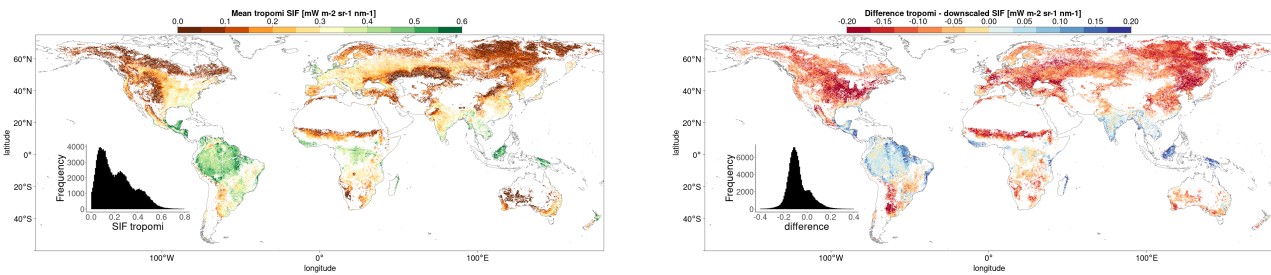

**Figure A5.** Left: The mean TROPOMI SIF in the 2020 growing season. Right: The difference (TROPOMI - downscaled) between mean TROPOMI SIF (2020) and mean downscaled SIF (2007-2014).

### A4.3  Intra-annual correlation between TROPOMI SIF and FLUXCOM GPP

Figure A6 displays the intra-annual correlation between TROPOMI SIF and FLUXCOM GPP, as well as the difference (TROPOMI - downscaled) when compared to the intra-annual correlation between downscaled SIF and FLUXCOM GPP. The figure shows that across the growing season, the intra-annual correlation between TROPOMI SIF and FLUXCOM GPP is very similar to that of the downscaled SIF and GPP across a growing season, with the vast majority of points showing a difference of less that 0.1. Significant differences between the two SIF products are mostly observed in equatorial rainforests.

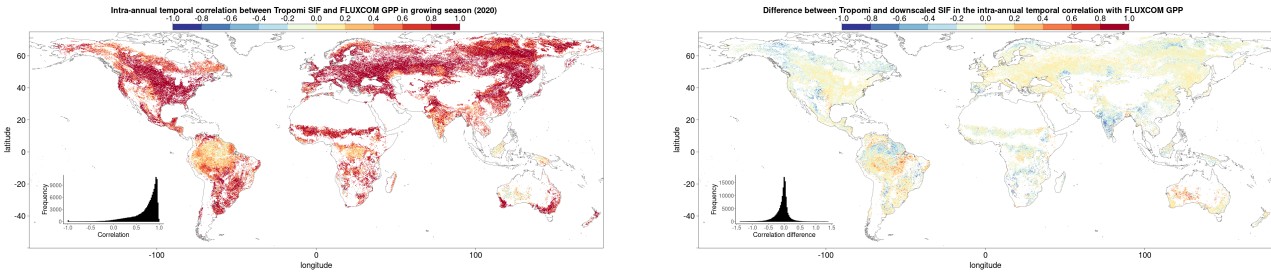

**Figure A6.** Left: The intra-annual temporal correlation between TROPOMI SIF and FLUXCOM GPP in 2020. Right: The difference (TROPOMI - downscaled) between TROPOMI SIF (2020) and downscaled SIF (2007-2014) in the intra-annual temporal correlation between SIF and FLUXCOM GPP.

### A4.4  Spatial relationship between TROPOMI SIF and FLUXCOM GPP

Figure A7 shows the relative distribution and spatial linear relationship between the mean growing season FLUXCOM GPP as a function of the respective mean values of the TROPOMI SIF. The data are broken down into separate categories depending on the Köppen-Geiger climate grouping and dominant vegetation cover of the pixel. The figures show spatial correlations for the TROPOMI SIF and FLUXCOM GPP, that are broadly similar with those of the downscaled SIF and FLUXCOM GPP.

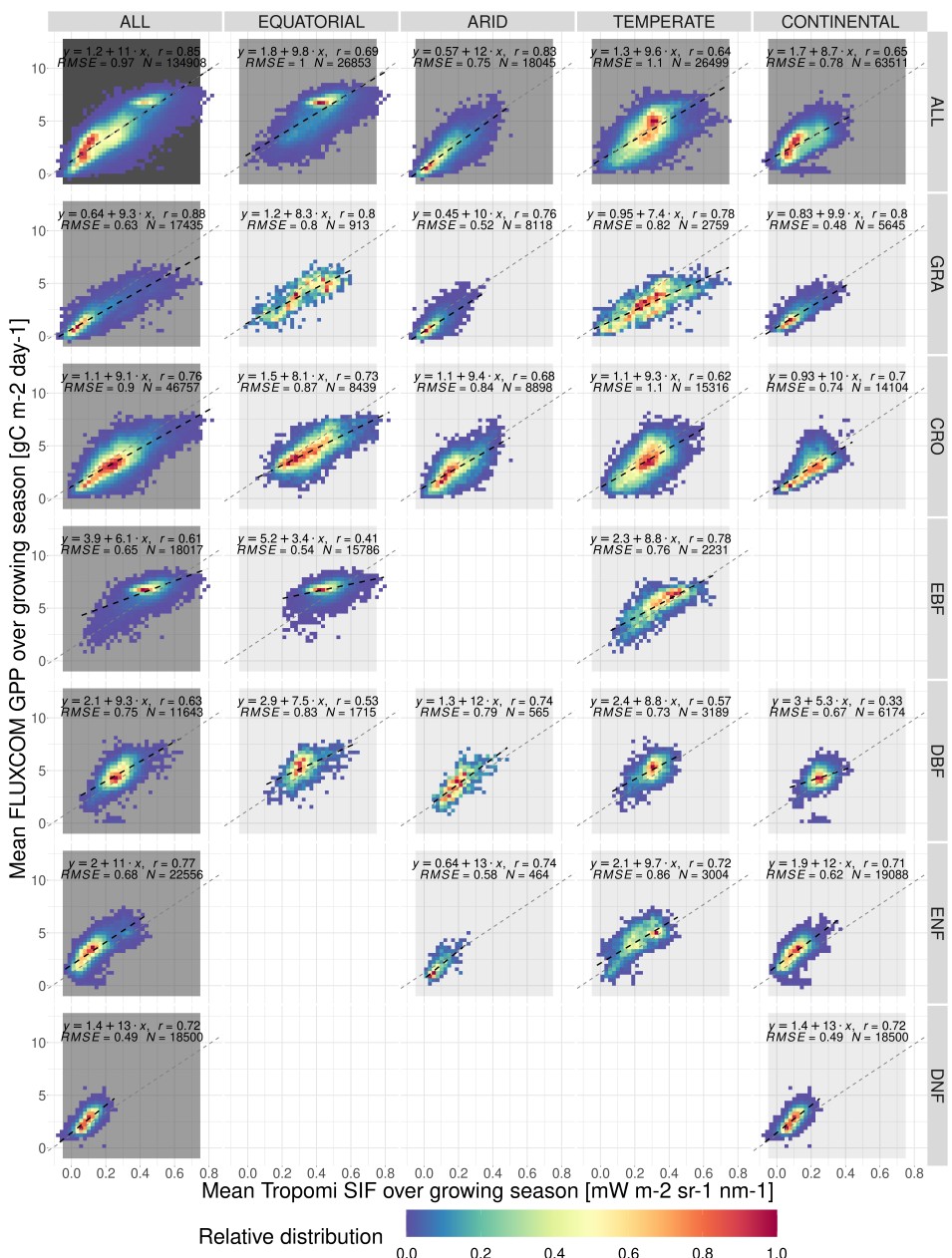

**Figure A7.** The spatial relationship between the mean growing season TROPOMI SIF and FLUXCOM GPP, broken down into separate Köppen-Geiger climate zones and vegetation cover categories. The plot shows the frequency distribution of pixels into $SIF_{DS}$-$GPP_{FX}$ bins, relative to the highest frequency bin in that category. A black dashed line representing a linear model in each category is overlaid and compared to a grey dotted line representing a linear model produced without the breakdown into separate categories (i.e. 'ALL-ALL'). The linear model equation, correlation coefficient $r$, root mean squared error (RMSE) and number of pixels are included.

## A4.5 Spatial analysis of covariance between TROPOMI SIF and FLUXCOM GPP

The ANCOVA analysis is repeated for the TROPOMI SIF with FLUXCOM GPP for the full year of 2020, and the $\eta^2$ parameters for slope and intercept are displayed in figure A8, whilst the full results can be found in table A3.

The results support the main features of the analysis with downscaled SIF seen in figure 6. The difference in the scaling of the SIF-GPP relationship (i.e. the slope) between vegetation covers is relatively unimportant. There are however, a few exceptions, the most significant of which is evergreen broadleaf forests in equatorial regions. The difference between deciduous broadleaf forests and other vegetation covers in temperate regions is no longer present, suggesting it may be a feature of the downscaled SIF. There is a slight distinction that can be drawn between the scaling of continental needleleaf forests and other vegetation covers. In general, there is a slight decrease in the differences between the vegetation covers in the TROPOMI SIF dataset.

Though the slopes are similar, a reasonable proportion of the difference in the intercept of the linear relationship is attributable to the difference in vegetation covers. This difference broadly divides along the lines of herbaceous or non-woody vegetation (CRO, GRA) and woody vegetation (EBF, DBF, ENF, DNF). The intercept, which can be interpreted as the starting potential of the SIF-GPP relationship, is generally higher for woody trees (i.e. more SIF is released for a given GPP).

*Author contributions.* **Mark Pickering:** Conceptualisation, Methodology, Software, Validation, Formal analysis, Investigation, Visualization, Resources, Data curation, Writing–original draft, Writing–review and editing. **Alessandro Cescatti:** Conceptualisation, Resources, Writing–review and editing, Supervision, Funding acquisition. **Gregory Duveiller:** Conceptualisation, Methodology, Resources, Writing–review and editing, Supervision

*Competing interests.* The authors declare that they have no conflict of interest.

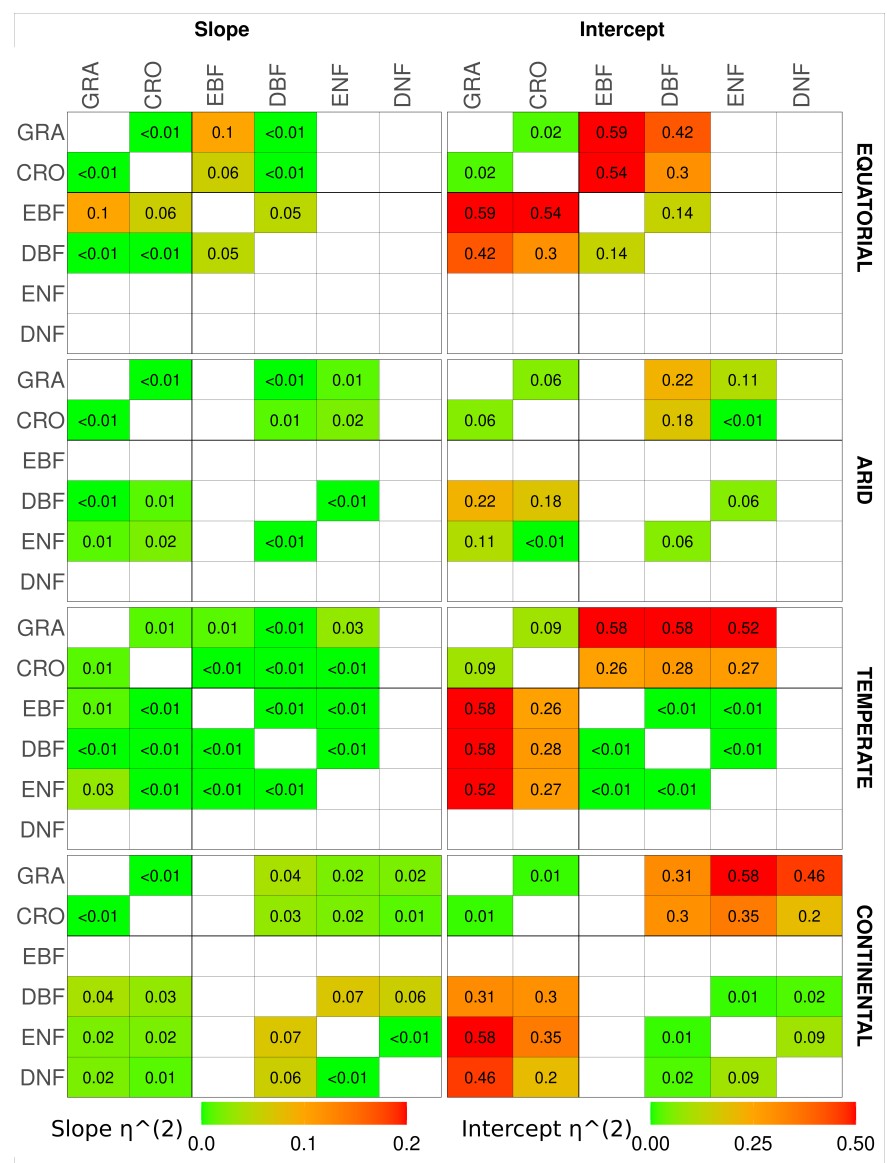

**Figure A8.** The $\eta^2$ parameter of an analysis of covariance between pairs of vegetation covers in different Köppen-Geiger climate groupings, for the slope (left) and intercept (right) of the linear relationship between TROPOMI SIF and FLUXCOM GPP. ANCOVA is performed on the intercept under the assumption that the difference between slopes is not significant. The $\eta^2$ parameter is comparable to the percentage of the difference in the slope or intercept (the latter assuming equivalence of the slopes) attributable to the difference in vegetation cover, with lower values signifying a smaller difference between vegetation covers. A slightly bolder line is used to separate the herbaceous species (CRO, GRA) from the woody species (EBF, DBF, ENF, DNF).

| Climate | land cover | | slope | | intercept | |
|---|---|---|---|---|---|---|
| | $LC_1$ | $LC_2$ | p-value | $\eta^2$ | p-value | $\eta^2$ |
| EQUATORIAL | EBF | DBF | $1.13 \times 10^{-22}$ | 0.05 | $5.77 \times 10^{-65}$ | 0.14 |
| EQUATORIAL | GRA | EBF | $2.68 \times 10^{-43}$ | 0.10 | $< 1.00 \times 10^{-99}$ | 0.59 |
| EQUATORIAL | CRO | EBF | $1.26 \times 10^{-30}$ | 0.06 | $< 1.00 \times 10^{-99}$ | 0.54 |
| EQUATORIAL | CRO | DBF | $7.52 \times 10^{-01}$ | $< 0.01$ | $< 1.00 \times 10^{-99}$ | 0.30 |
| EQUATORIAL | GRA | DBF | $1.09 \times 10^{-01}$ | $< 0.01$ | $< 1.00 \times 10^{-99}$ | 0.42 |
| EQUATORIAL | GRA | CRO | $9.00 \times 10^{-02}$ | $< 0.01$ | $1.16 \times 10^{-08}$ | 0.02 |
| ARID | GRA | DBF | $5.00 \times 10^{-03}$ | $< 0.01$ | $7.84 \times 10^{-88}$ | 0.22 |
| ARID | CRO | DBF | $1.58 \times 10^{-05}$ | 0.01 | $1.98 \times 10^{-68}$ | 0.18 |
| ARID | DBF | ENF | $8.00 \times 10^{-02}$ | $< 0.01$ | $4.98 \times 10^{-16}$ | 0.06 |
| ARID | GRA | CRO | $1.90 \times 10^{-02}$ | $< 0.01$ | $1.60 \times 10^{-27}$ | 0.06 |
| ARID | GRA | ENF | $2.85 \times 10^{-06}$ | 0.01 | $2.53 \times 10^{-39}$ | 0.11 |
| ARID | CRO | ENF | $1.34 \times 10^{-06}$ | 0.02 | $4.50 \times 10^{-01}$ | $< 0.01$ |
| TEMPERATE | DBF | ENF | $1.00 \times 10^{-03}$ | $< 0.01$ | $9.40 \times 10^{-01}$ | $< 0.01$ |
| TEMPERATE | GRA | DBF | $4.20 \times 10^{-02}$ | $< 0.01$ | $< 1.00 \times 10^{-99}$ | 0.58 |
| TEMPERATE | CRO | DBF | $7.10 \times 10^{-02}$ | $< 0.01$ | $< 1.00 \times 10^{-99}$ | 0.28 |
| TEMPERATE | EBF | DBF | $1.60 \times 10^{-01}$ | $< 0.01$ | $9.44 \times 10^{-01}$ | $< 0.01$ |
| TEMPERATE | GRA | ENF | $1.66 \times 10^{-13}$ | 0.03 | $< 1.00 \times 10^{-99}$ | 0.52 |
| TEMPERATE | CRO | ENF | $2.30 \times 10^{-01}$ | $< 0.01$ | $< 1.00 \times 10^{-99}$ | 0.27 |
| TEMPERATE | EBF | ENF | $6.00 \times 10^{-03}$ | $< 0.01$ | $6.58 \times 10^{-01}$ | $< 0.01$ |
| TEMPERATE | GRA | EBF | $3.53 \times 10^{-08}$ | 0.01 | $< 1.00 \times 10^{-99}$ | 0.58 |
| TEMPERATE | CRO | EBF | $3.12 \times 10^{-01}$ | $< 0.01$ | $< 1.00 \times 10^{-99}$ | 0.26 |
| TEMPERATE | GRA | CRO | $4.58 \times 10^{-07}$ | 0.01 | $1.85 \times 10^{-44}$ | 0.09 |
| CONTINENTAL | DBF | DNF | $2.03 \times 10^{-29}$ | 0.06 | $9.89 \times 10^{-10}$ | 0.02 |
| CONTINENTAL | GRA | DNF | $4.91 \times 10^{-11}$ | 0.02 | $< 1.00 \times 10^{-99}$ | 0.46 |
| CONTINENTAL | DBF | ENF | $5.96 \times 10^{-32}$ | 0.07 | $1.55 \times 10^{-04}$ | 0.01 |
| CONTINENTAL | GRA | ENF | $3.91 \times 10^{-12}$ | 0.02 | $< 1.00 \times 10^{-99}$ | 0.58 |
| CONTINENTAL | CRO | DNF | $6.18 \times 10^{-08}$ | 0.01 | $1.69 \times 10^{-96}$ | 0.20 |
| CONTINENTAL | CRO | DBF | $1.61 \times 10^{-12}$ | 0.03 | $< 1.00 \times 10^{-99}$ | 0.30 |
| CONTINENTAL | CRO | ENF | $1.16 \times 10^{-09}$ | 0.02 | $< 1.00 \times 10^{-99}$ | 0.35 |
| CONTINENTAL | GRA | DBF | $2.58 \times 10^{-18}$ | 0.04 | $< 1.00 \times 10^{-99}$ | 0.31 |
| CONTINENTAL | GRA | CRO | $6.82 \times 10^{-01}$ | $< 0.01$ | $8.96 \times 10^{-05}$ | 0.01 |
| CONTINENTAL | ENF | DNF | $9.00 \times 10^{-01}$ | $< 0.01$ | $3.40 \times 10^{-44}$ | 0.09 |

**Table A3.** Analysis of covariance between pairs of land covers in different Köppen-Geiger climate groupings for the relationship between TROPOMI SIF and FLUXCOM GPP. ANCOVA is only performed on the intercept when the difference between slopes is not considered significant.

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
