# Peer review of "Sun-Induced Fluorescence as a Proxy of Primary Productivity across Vegetation Types and Climates"

_Biogeosciences, 2021_

## Author Response (AR1)

Thank you for the important comments, and suggestions for improving the manuscript. In light of the most recent comments, we decided to revisit the manuscript and rerun the analysis with a year of SIF data from tropomi (compared with FLUXCOM GPP) and comparing the downscaled SIF with GPP from FluxSat. The comparisons can be found in the appendix and support the main conclusions of the paper, particularly with regard to differences in the SIF-GPP scaling between land covers. We believe these revisions provide important and useful evidence in support of the publication of the paper.

We provide differences (i.e. latexdiff version_previous.txt version_latest.txt) to aide the reviewers and we also draw attention to the follow changes:

1. Rather than displaying the results of the analysis of covariance in table format, we instead provide a figure. This allows for a continuous colour scale that is easier to compare (e.g. with other data sources). The full tables seen previously are provided in the appendix. The phrasing of the relevant sections have also been adjusted accordingly.

2. The addition of a section re-running some of the spatial analyses between SIF and GPP using downscaled SIF and an alternative GPP dataset: FluxSat. This involves
    1. A comparison of the FluxSat GPP and FLUXCOM GPP distribution
    2. The spatial linear relationships between downscaled SIF and FluxSat GPP for different vegetation covers
    3. An analysis of the covariance between downscaled SIF and FluxSat GPP

3. The addition of a section re-running most of the comparative analyses between SIF and GPP using the Tropomi SIF (for the year 2020) in place of the downscaled SIF. This involves
    1. A comparison of the Tropomi SIF and downscaled SIF distribution
    2. The intra-annual tropomi SIF - FLUXCOM GPP correlation
    3. The spatial linear relationships between tropomi SIF and FLUXCOM GPP for different vegetation covers
    4. An analysis of covariance between Tropomi SIF and FLUXCOM GPP

4. There is some discussion within the text of the fact that the tropomi and FluxSat appendix results support the conclusions drawn from the downscaled SIF.
    1. This is mentioned in the abstract
        1. 'Additional analyses with alternative SIF and GPP datasets support these conclusions.'
    2. The end of the introduction
        1. 'Similarly, comparisons with alternative SIF and GPP products such as Tropomi SIF and FluxSat GPP are  provided in an appendix, in order to

ensure the consistency and robustness of the conclusions (Joiner and Yoshida, 2021; Köhler et al., 2018b).'

3. The methodology (section 3)
    1. Several sections of the analysis of the SIF-GPP spatio-temporal relationship are repeated with the alternative FluxSat GPP dataset (in place of the FLUXCOM GPP) and the Tropomi SIF dataset (in place of the downscaled SIF) in order to ensure the robustness and consistency of the analysis. These can be found in appendix A3 and appendix A4 respectively.

4. The end of the spatial linear relationship section 4.2
    1. There is a reference to the SIF-GPP spatial correlation being stronger in the fluxsat dataset
    2. Also the mention that a feature in the downscaled SIF in temperate DBF regions is reduced in the tropomi SIF

5. The ANCOVA results section 4.3
    1. Appendix A1 contains the full table of results, whilst similar analyses comparing the downscaled SIF - FluxSat GPP relationship and the Tropomi SIF - FLUXCOM GPP relationship can be found in appendices A3 and A4 respectively

6. The discussion
    1. 5.1 utility of downscaled SIF
        1. The reproduction of known SIF-GPP patterns using the downscaled SIF demonstrates its utility as a high-resolution proxy of primary productivity. In support of these conclusions, appendix A4 replicates the main analysis results with the substitution of a single year of Tropomi data in place of the downscaled SIF, whilst appendix A3 ensures the conclusions are not unique to the choice of the GPP dataset. In this sense the analysis serves as a diagnostic benchmark for the comparison of SIF and GPP datasets.

All other changes can be found in the discussion here: https://bg.copernicus.org/preprints/bg-2021-354/#discussion

---

## Author Response (AR2)

Dear All,

Thank you for your accepting this manuscript to the journal. The comments and suggestions were helpful in improving the manuscript and giving confidence in the results, particularly with regards to the addition of alternative datasets in the appendix.

There are no additional changes to this version with the exception of conversion to the required submission format and latex template (including the addition of author contributions and data availability), and with the correction of a typo in figure 5 (Tropomi SIF -> downscaled SIF).

We hope the submission is in the correct format and standard.

Kind regards,